# Simulation-based survey of TMEM16 family reveals that robust lipid scrambling requires an open groove

Christina Alexandra Stephens[1,2,3†], Niek van Hilten[1,3†], Lisa Zheng[1,2,3†], Michael Grabe[1]*

[1]Cardiovascular Research Institute, University of California, San Francisco, San Francisco, United States; [2]Graduate Group in Biophysics, University of California, San Francisco, San Francisco, United States; [3]Department of Pharmaceutical Chemistry, University of California, San Francisco, San Francisco, United States

*For correspondence:
michael.grabe@ucsf.edu

†These authors contributed equally to this work

Competing interest: The authors declare that no competing interests exist.

## eLife Assessment

This **important** study provides information on the TMEM16 family of membrane proteins, which play roles in lipid scrambling and ion transport. By simulating 27 structures representing five distinct family members, the authors captured hundreds of lipid scrambling events, offering insights into the mechanisms of lipid translocation and the specific protein regions involved in these processes. While the data on comparison of scrambling competence is **compelling**, the evidence for outside-the-groove scramblase activity without experimental validation is missing and is based on a limited set of observed events.

**Abstract** Biological membranes are complex and dynamic structures with different populations of lipids in their inner and outer leaflets. The $Ca^{2+}$-activated TMEM16 family of membrane proteins plays an important role in collapsing this asymmetric lipid distribution by spontaneously and bidirectionally scrambling phospholipids between the two leaflets, which can initiate signaling and alter the physical properties of the membrane. While evidence shows that lipid scrambling can occur via an open hydrophilic pathway (groove) that spans the membrane, it remains unclear if all family members facilitate lipid movement in this manner. Here, we present a comprehensive computational study of lipid scrambling by all TMEM16 members with experimentally solved structures. We performed coarse-grained molecular dynamics (MD) simulations of 27 structures from five different family members solved under activating and non-activating conditions, and we captured over 700 scrambling events in aggregate. This enabled us to directly compare scrambling rates, mechanisms, and protein–lipid interactions for fungal and mammalian TMEM16s, in both open ($Ca^{2+}$-bound) and closed ($Ca^{2+}$-free) conformations with statistical rigor. We show that all TMEM16 structures thin the membrane and that the majority of scrambling (>90%) occurs at the groove only when TM4 and TM6 have sufficiently separated. Surprisingly, we also observed 60 scrambling events that occurred outside the canonical groove, over 90% of which took place at the dimer–dimer interface in mammalian TMEM16s. This new site suggests an alternative mechanism for lipid scrambling in the absence of an open groove.

## Introduction

The TMEM16 family of eukaryotic membrane proteins, also known as anoctamins, is comprised of lipid scramblases (*Suzuki et al., 2013*; *Di Zanni et al., 2018*; *Kim et al., 2018*), ion channels (*Yang*

*et al., 2008*; *Caputo et al., 2008*; *Stöhr et al., 2009*; *Schroeder et al., 2008*), and members that can facilitate both lipid and ion permeation (*Yang et al., 2012*; *Martins et al., 2011*; *Suzuki et al., 2010*; *Malvezzi et al., 2013*; *Bushell et al., 2019*; *Lee et al., 2016*; *Scudieri et al., 2015*; *Falzone et al., 2019*). This functional divergence, despite their high sequence conservation, is a unique feature among the 10 vertebrate paralogs (*Milenkovic et al., 2010*). So far, all characterized TMEM16s require $Ca^{2+}$ to achieve their maximum transport activity, whether that be passive ion movement or lipid flow down their electrochemical gradients (*Yang et al., 2012*; *Malvezzi et al., 2013*; *Terashima et al., 2013*; *Yu et al., 2012*; *Brunner et al., 2014*; *Hartzell et al., 2009*; *Alvadia et al., 2019*). TMEM16s play critical roles in a variety of physiological processes including blood coagulation (*Yang et al., 2012*; *Castoldi et al., 2011*; *Boisseau et al., 2018*; *Brooks et al., 2015*), bone mineralization (*Tsutsumi et al., 2004*), mucus secretion (*Cabrita et al., 2021*), smooth muscle contraction (*Huang et al., 2012*), and membrane fusion (*Griffin et al., 2016*). Mutations of TMEM16 have also been implicated in several cancers (*Mohsenzadegan et al., 2013*; *Chen et al., 2021*; *Cereda et al., 2010*), neuronal disorder SCAR10 (*Vermeer et al., 2010*; *Petkovic et al., 2019*), and SARS-CoV-2 infection (*Braga et al., 2021*). Despite their significant roles in human physiology, the functional properties of most vertebrate TMEM16 paralogs remain unknown. Moreover, even though we have significant functional and structural insight into the mechanisms of a handful of members (*Malvezzi et al., 2013*; *Lee et al., 2016*; *Falzone et al., 2019*; *Stansfeld et al., 2015*; *Bethel and Grabe, 2016*; *Jiang et al., 2017*; *Khelashvili et al., 2020*; *Khelashvili et al., 2019*; *Lee et al., 2018*; *Cheng et al., 2022*; *Kostritskii and Machtens, 2021*; *Malvezzi et al., 2018*; *Peters et al., 2015*; *Peters et al., 2018*; *Paulino et al., 2017*; *Le et al., 2019a*; *Jia and Chen, 2021*; *Lam et al., 2022*; *Yu et al., 2019*; *Lowry et al., 2024*; *Le et al., 2019b*; *Kmit et al., 2013*; *Han et al., 2019*; *Khelashvili et al., 2022*; *Segawa et al., 2011*; *Watanabe et al., 2018*; *Jia et al., 2022*), it is still an open question whether all TMEM16s work in the same way to conduct ions or scramble lipids.

Over the 10 years, 63 experimental structures of TMEM16s have been determined, revealing a remarkable structural similarity between mammalian and fungal members despite the diversity in their functions. All structures, except for one of fungal *Aspergillus fumigatus* TMEM16 (afTMEM16) (*Falzone et al., 2022*), which is a monomer, are homodimers with a butterfly-like fold (*Bushell et al., 2019*; *Falzone et al., 2019*; *Brunner et al., 2014*; *Alvadia et al., 2019*; *Paulino et al., 2017*; *Lam et al., 2022*; *Falzone et al., 2022*; *Arndt et al., 2022*; *Kalienkova et al., 2019*; *Feng et al., 2024a*; *Dang et al., 2017*; *Lam and Dutzler, 2021*; *Lam and Dutzler, 2023*; *Feng et al., 2019*; *Feng et al., 2023*), and each subunit is comprised of 10 transmembrane (TM) helices with the final helix (TM10) forming most of the dimer interface. Residues on TM6 form half of a highly conserved $Ca^{2+}$-binding site that accommodates up to two ions. TM6, along with TM3, TM4, and TM5, also forms a membrane spanning groove that contains hydrophilic residues that are shielded from the hydrophobic core of the bilayer in $Ca^{2+}$-free states. When $Ca^{2+}$ is bound, TM6 takes on a variety of conformational and secondary structural changes across the family, which can have profound effects on the shape of the membrane as seen in cryo-EM nanodiscs with TMEM16F (*Feng et al., 2019*). $Ca^{2+}$ binding is also associated with the movement of the upper portion of TM4 away from TM6 which effectively exposes (opens) the hydrophilic groove to the bilayer, but this opening is not observed for all $Ca^{2+}$-bound TMEM16 members (*Alvadia et al., 2019*; *Paulino et al., 2017*; *Lam et al., 2022*; *Kalienkova et al., 2019*; *Dang et al., 2017*; *Lam and Dutzler, 2021*; *Lam and Dutzler, 2023*; *Feng et al., 2019*; *Feng et al., 2023*).

It was first theorized (*Brunner et al., 2014*; *Yu et al., 2015*) and later predicted by molecular dynamics (MD) simulations (*Bushell et al., 2019*; *Stansfeld et al., 2015*; *Bethel and Grabe, 2016*; *Lee et al., 2018*; *Le et al., 2019b*; *Khelashvili et al., 2022*) that lipids can traverse the membrane bilayer by moving their headgroups along the water-filled hydrophilic groove (between TM4 and TM6) while their tails project into the hydrophobic center of the bilayer. This mechanism for scrambling, first proposed by Menon and Pomorski, is often referred to as the 'credit card model' (*Pomorski and Menon, 2006*). All-atom MD (AAMD) simulations of open *Nectria haematococca* TMEM16 (nhTMEM16) have shown that lipids near the pore frequently interact with charged residues at the groove entrances (*Bethel and Grabe, 2016*), two of which are in the scrambling domain which confers scramblase activity to the ion channel-only member TMEM16A (*Gyobu et al., 2017*). Frequent headgroup interactions with residues lining the groove were also noted in atomistic simulations of open TMEM16K including two basic residues in the scrambling domain. Lipids experience a relatively low

energy barrier for scrambling in open nhTMEM16 (<1 kcal/mol compared to 20–50 kcal/mol directly through the bilayer) (*Bethel and Grabe, 2016*; *Pomorski and Menon, 2006*). Simulations also indicate that zwitterionic lipid headgroups stack in the open groove along their dipoles, which may help energetically stabilize them during scrambling (*Bethel and Grabe, 2016*; *Kostritskii and Machtens, 2021*). Finally, simulations also show that lipids can directly gate nhTMEM16 groove opening and closing through interactions with their headgroups or tails (*Khelashvili et al., 2020*; *Khelashvili et al., 2019*). It is important to note that all of these simulation observations are based on a limited number of spontaneous events from different groups (in aggregate, we estimate that no more than 14 scrambling events have been reported in the absence of an applied voltage) (*Bushell et al., 2019*; *Bethel and Grabe, 2016*; *Jiang et al., 2017*; *Lee et al., 2018*; *Le et al., 2019b*; *Khelashvili et al., 2022*). Many more scrambling events (~800 in aggregate) have been seen in coarse-grained MD (CGMD) simulations for nhTMEM16 (*Stansfeld et al., 2015*), TMEM16K (*Bushell et al., 2019*), mutant TMEM16F (F518H), and even TMEM16A (*Li et al., 2024*); however, a detailed analysis of how scrambling occurred in these latter two was not provided. Moreover, a head-to-head comparison of fungal versus mammalian scrambling rates has not been made.

An outstanding question in the field is whether scrambling requires an open groove. This question has been triggered in part by the failure to determine wild-type (WT) TMEM16F structures with open grooves wide enough to accommodate lipids, despite structures being solved under activating conditions (*Feng et al., 2019*; *Feng et al., 2023*). Further uncertainty stems from data showing that scrambling can occur in the absence of $Ca^{2+}$ when the groove is presumably closed (*Malvezzi et al., 2013*; *Bushell et al., 2019*; *Lee et al., 2016*; *Brunner et al., 2014*; *Lee et al., 2018*; *Malvezzi et al., 2018*; *Feng et al., 2024a*). Moreover, afTMEM16 can scramble PEGylated lipids, which are too large for even the open groove (*Malvezzi et al., 2018*). This last finding motivated Malvezzi et al. to propose an alternate model of scrambling (*Malvezzi et al., 2018*) inspired by the realization that the bilayer adjacent to the protein, whether the groove is open or closed, is distorted in cryo-EM in nanodiscs (*Falzone et al., 2022*; *Arndt et al., 2022*; *Kalienkova et al., 2019*; *Feng et al., 2019*), MD simulations (*Bushell et al., 2019*; *Bethel and Grabe, 2016*; *Khelashvili et al., 2020*), and continuum models (*Bethel and Grabe, 2016*). The hallmark of this distortion is local bending and thinning adjacent to the groove (estimated to be 50–60% thinner than bulk for some family members) (*Bushell et al., 2019*; *Bethel and Grabe, 2016*; *Khelashvili et al., 2020*; *Falzone et al., 2022*; *Arndt et al., 2022*; *Kalienkova et al., 2019*; *Feng et al., 2019*), and it has been suggested that this deformation, along with packing defects, may significantly lower the energy barrier for lipid crossing (*Bethel and Grabe, 2016*; *Falzone et al., 2022*; *Feng et al., 2019*). To date, no AAMD or CGMD simulation has reported scrambling by any WT TMEM16 harboring a closed groove; however, a CGMD simulation of the F518H TMEM16F mutant did report scrambling, but the details, such as whether the groove opened, were not provided (*Li et al., 2024*). Again, since a comprehensive analysis across all family members has not been carried out, it is difficult to determine how membrane thinning is related to scrambling or if scrambling mechanisms are specific to certain family members, conformational states of the protein, or both. Additionally, lipids are also directly involved in how TMEM16 scramblases conduct ions. As first speculated in *Malvezzi et al., 2013*, AAMD simulations have shown that ions permeate through the lipid headgroup-lined hydrophilic groove of TMEM16K and nhTMEM16 (*Jiang et al., 2017*; *Khelashvili et al., 2019*; *Cheng et al., 2022*; *Kostritskii and Machtens, 2021*; *Jia et al., 2022*). How might this mechanism differ in the absence of an open groove?

To address these outstanding questions, we employed CGMD simulation to systematically quantify scrambling in 23 experimental and 4 computationally predicted TMEM16 proteins taken from each family member that has been structurally characterized: fungal scramblases nhTMEM16 and afTMEM16, and mammalian scramblase TMEM16K, TMEM16F, and TMEM16A (*Appendix 1—table 1*; *Appendix 1—figure 1*). CGMD, which was the first computational method to identify nhTMEM16 as a scramblase (*Stansfeld et al., 2015*), enables us to reach much longer time scales, while retaining enough chemical detail to faithfully reproduce experimentally verified protein–lipid interactions (*Moss et al., 2023*). This allowed us to quantitatively compare the scrambling statistics and mechanisms of different WT and mutant TMEM16s in both open and closed states solved under different conditions (e.g., salt concentrations, lipid and detergent environments, in the presence of modulators or activators like $PIP_2$ and $Ca^{2+}$). Our simulations successfully reproduce experimentally determined membrane deformations seen in nanodiscs across both fungal and mammalian TMEM16s. They also show that

only open scramblase structures have grooves fully lined by lipids, and each of these structures promotes scrambling in the groove with lipids experiencing a less than 1 kT free energy barrier as they move between leaflets. Interestingly, one simulation of TMEM16A, which is not a scramblase, initiated from a predicted ion-conductive state scrambled lipids through a lipid-lined groove at a very low rate (only two events), suggesting that ion channel-only members may have residual non-detectable scramblase activity. Our analysis of the membrane deformation and groove conformation shows that most scrambling in the groove occurs when the membrane is thinned to at least 14 Å and the groove is open. We also observe 218 ion permeation events but only in well-hydrated systems with open grooves (98%) and a closed-groove TMEM16A structure (2%). Our simulations also reveal alternative scrambling pathways, which primarily occur at the dimer–dimer interface in mammalian structures.

## Results
### Lipid densities from coarse-grained simulations match all-atom simulations and cryo-EM nanodiscs

We simulated $Ca^{2+}$-bound and -free (apo) structures of TMEM16 proteins in a 1,2-dioleoyl-sn-glycero-3-phosphatidylcholine (DOPC) bilayer for 10 µs each using the Martini 3 force field. First, we determined how well the simulated membrane distortions matched experiment by comparing the annulus of lipids surrounding each protein to the lipid densities derived from structures solved in nanodiscs (*Figure 1—figure supplement 1*). The shapes of the membrane near the protein qualitatively match the experimental densities and the shapes produced from AAMD simulations and continuum membrane models (*Bethel and Grabe, 2016*; *Falzone et al., 2022*; *Kalienkova et al., 2019*). For example, the CG simulations capture the sinusoidal curve around both fungal scramblases in apo and $Ca^{2+}$-bound states (*Figure 1—figure supplement 1*) previously determined by atomistic simulations (*Bethel and Grabe, 2016*) of $Ca^{2+}$-bound nhTMEM16 (*Figure 1A, B*). Even though membrane deformation is a general feature of TMEM16s, the shapes between fungal and mammalian members are noticeably different. Specifically, the membrane is flatter around TMEM16K and TMEM16F compared to the fungal members in both the nanodisc density and CGMD (*Figure 1—figure supplement 1*). For WT TMEM16s, whether the groove is open (*Figure 1B*, *insets*) or closed (*Figure 1—figure supplement 2*), strong lipid density exists near the extracellular groove entrances at TM1 and TM8. Interestingly, this density is lost in the simulation of the $Ca^{2+}$-bound constitutively active TMEM16F F518H mutant (PDB ID 8B8J), consistent with what is seen in the cryo-EM structure solved in nanodisc (*Figure 1—figure supplement 1*). The lipid density is present, however, at this location for the simulated open $Ca^{2+}$-bound WT TMEM16F (6QP6*, initiated from PDB ID 6QP6) and closed $Ca^{2+}$-bound WT TMEM16F (PDB ID 6QP6) (*Figure 1C*). The loss of density indicates that the normal membrane contact with the protein near the TMEM16F groove has been compromised in the mutant structure. Residues in this site on nhTMEM16 and TMEM16F also seem to play a role in scrambling, but the mechanism by which they do so is unclear (*Falzone et al., 2022*; *Feng et al., 2024a*; *Feng et al., 2023*).

Headgroup density isosurfaces from CGMD simulations of known scramblases bound to $Ca^{2+}$ and with clear separation of TM4 and TM6 show that lipid headgroups occupy the full length of the groove creating a clear pathway that links the upper and lower membrane leaflets (nhTMEM16 (PDB ID 4WIS), afTMEM16 (PDB ID 7RXG), TMEM16K (PDB ID 5OC9) and TMEM16F F518H (PDB ID 8B8J), *Figure 1B*). These simulation-derived densities crossing the bilayer are strikingly similar to lipids resolved in cryo-EM structures of fungal scramblases in nanodisc (*Falzone et al., 2022*; *Feng et al., 2024a*). Individual simulation snapshots provide insight into how lipids traverse this pathway. Open-groove structures typically have an average of 2–3 lipids in the TM4/TM6 groove at any given time (see *Figure 1—source data 1*). Additional analysis shows that all of the grooves are filled with water (*Appendix 1—figure 2*). These profiles share additional features including a clear upward deflection of the membrane as it approaches TM3/TM4 from the left and a downward deflection as it approaches TM6/TM8 from the right; however, the degree of this deflection is not equal, as can be seen for TMEM16K, which is less pronounced (*Figure 1B*). These distortions are coupled to the sinusoidal curve around the entire protein, which was shown to thin the membrane across the groove and hypothesized to aid in scrambling (*Bethel and Grabe, 2016*).

Unlike the open $Ca^{2+}$-bound scramblase structures, apo and closed $Ca^{2+}$-bound TMEM16 structures lack lipid headgroup density spanning the bilayer, and their density profiles are more consistent

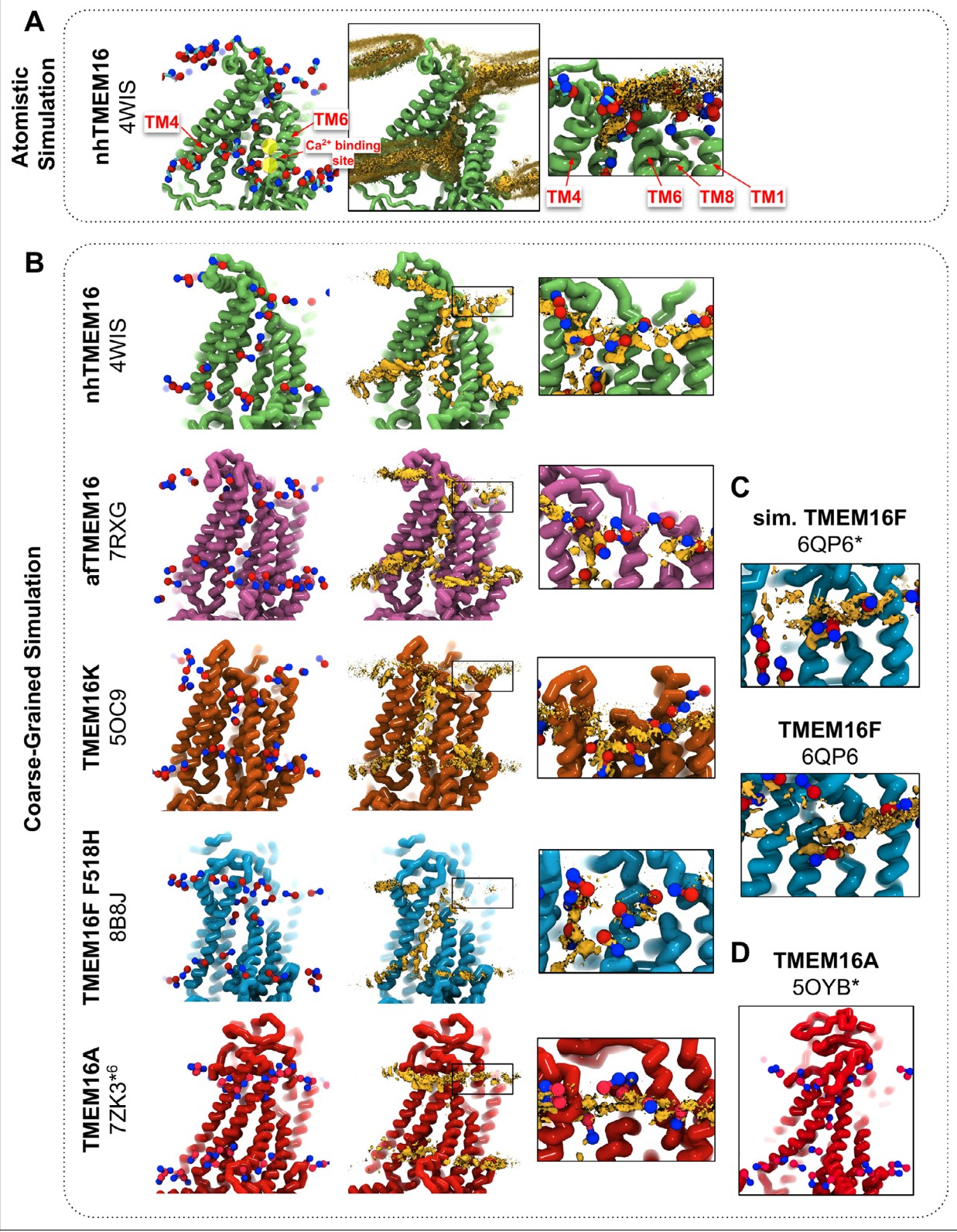

**Figure 1.** CG simulations of multiple TMEM16 structures capture lipid density in the TM4/TM6 pathway of scrambling-competent members. (**A**) Snapshot and POPC headgroup density (right) from atomistic simulations of Ca²⁺-bound nhTMEM16 (PDB ID 4WIS) previously published in *Bethel and Grabe, 2016*. Only the lipid headgroup choline (blue) and phosphate (red) beads are shown for clarity. Density (brown isosurface) is averaged from both subunits across eight independent simulations totaling ~2 μs. Two yellow circles indicate the Ca²⁺-binding sites. (**B**) Snapshots from CG simulations

*Figure 1 continued on next page*

*Figure 1 continued*

of open Ca²⁺-bound nhTMEM16 (PDB ID 4WIS, green), afTMEM16 (PDB ID 7RXG, violet), TMEM16K (PDB ID 5OC9, orange), TMEM16F F518H (PDB ID 8B8J, blue), TMEM16K (orange), and TMEM16A (red). (**C**) Snapshots with lipid headgroup densities near simulated open (6QP6*) and closed (PDB ID 6QP6) TMEM16F. (**D**) Snapshot of simulated ion-conductive TMEM16A (5OYB*). For each CG snapshot, again only the lipid headgroup choline (blue) and phosphate (red) beads are shown for clarity. Each density (brown isosurface) is averaged over both chains except TMEM16K and TMEM16A where only a single chain is used due to the structure's asymmetry.

The online version of this article includes the following source data and figure supplement(s) for figure 1:

**Source data 1.** The average number of lipids simultaneosly occupying the protein groove per simulation frame for all groove-scrambling competent structures.

**Figure supplement 1.** Comparison between membrane deformations in cryo-EM nanodiscs and coarse-grained molecular dynamics (CGMD).

**Figure supplement 2.** CG simulations of multiple TMEM16 structures with closed grooves lack lipid density in the TM4–TM6 pathway.

across the entire family (*Figure 1—figure supplement 2*). The membrane is deformed near the groove with some lower leaflet lipid density entering part of the groove and some of the upper leaflet density deflecting inward around TM1, TM6, and TM8 but not entering the closed outer portion of the groove. Again, the membrane around TMEM16F and TMEM16K is flatter than it is in the fungal scramblases. Similarly, simulation of a Ca²⁺-bound TMEM16A conformation that conducts Cl⁻ in AAMD (7ZK3*⁶, initiated from PDB ID 7ZK3, see Appendix 1—Methods and *Appendix 1—figure 1*) samples partial lipid headgroup penetration into the extracellular vestibule formed by TM3/TM6, but lipids fail to traverse the bilayer as indicated by the lack of density in the center of the membrane (*Figure 1B*). This finding is consistent with TMEM16A lacking scramblase activity (*Han et al., 2019*; *Gyobu et al., 2017*); however, we simulated another ion-conductive TMEM16A conformation that can achieve a fully lipid-lined groove during its simulation (*Figure 1D*), although this configuration was uncommon (*Figure 1—source data 1*).

## Simulations recapitulate scrambling competence of open and closed structures

To quantify the scrambling competence of each simulated TMEM16 structure, we determined the number of events in which lipids transitioned from one leaflet to the other (see Methods, *Figure 2—figure supplement 1*). The scrambling rates calculated from our MD trajectories are in excellent agreement with the presumed scrambling competence of each experimental structure (*Figure 2A*). The strongest scrambler was the open-groove, Ca²⁺-bound fungal nhTMEM16 (PDB ID 4WIS), with 24.4 ± 5.2 events per µs (*Figure 2—figure supplement 2A*). In line with experimental findings, the open Ca²⁺-free structure (PDB ID 6QM6), which is structurally very similar to PDB ID 4WIS, also scrambled lipids (15.7 ± 3.9 events per µs, *Figure 2—figure supplement 2B*; *Kalienkova et al., 2019*). In contrast, we observed no scrambling events for the intermediate- (PDB ID 6QMA) and closed- (PDB ID 6QM4, PDB ID 6QMB) groove nhTMEM16 structures. We observed a similar trend for the fungal afTMEM16, where our simulations identified the open Ca²⁺-bound cryo-EM structure (PDB ID 7RXG) as scrambling competent (10.7 ± 2.9 events per µs, *Figure 2—figure supplement 2C*) while the Ca²⁺-free closed-groove structure (PDB ID 7RXB) was not.

For TMEM16K, our simulations showed that the Ca²⁺-bound X-ray structure (PDB ID 5OC9) facilitates scrambling (8.2 ± 2.9 events per µs) in line with experiments in the presence of Ca²⁺, when the groove is presumably open, and previous MD simulations (*Bushell et al., 2019*). Interestingly, we found a stark asymmetry in the number of scrambling events between the two monomers, with >80% of events happening via chain B (*Figure 2—figure supplement 3A*). Although both monomers are Ca²⁺-bound, chain B has a slightly wider endoplasmic reticulum (ER) lumen-facing entrance to the groove in the experimental starting structure (PDB ID 5OC9, *Figure 2—figure supplement 3B*) and spontaneously opened its groove more than subunit A during the simulation (8.2 ± 1.3 Å compared to 5.8 ± 0.6 Å on average, *Figure 3—figure supplement 6A*), which likely accounts for the increased rate. The closed-groove TMEM16K conformation (PDB ID 6R7X) showed very little scrambling activity (0.4 ± 0.7 events per µs).

Although TMEM16F is a known lipid scramblase found in the plasma membrane of platelets (*Suzuki et al., 2010*), none of the WT structures solved to date, even those determined under activating conditions, have exhibited an open hydrophilic groove. We simulated 10 of these proteins and

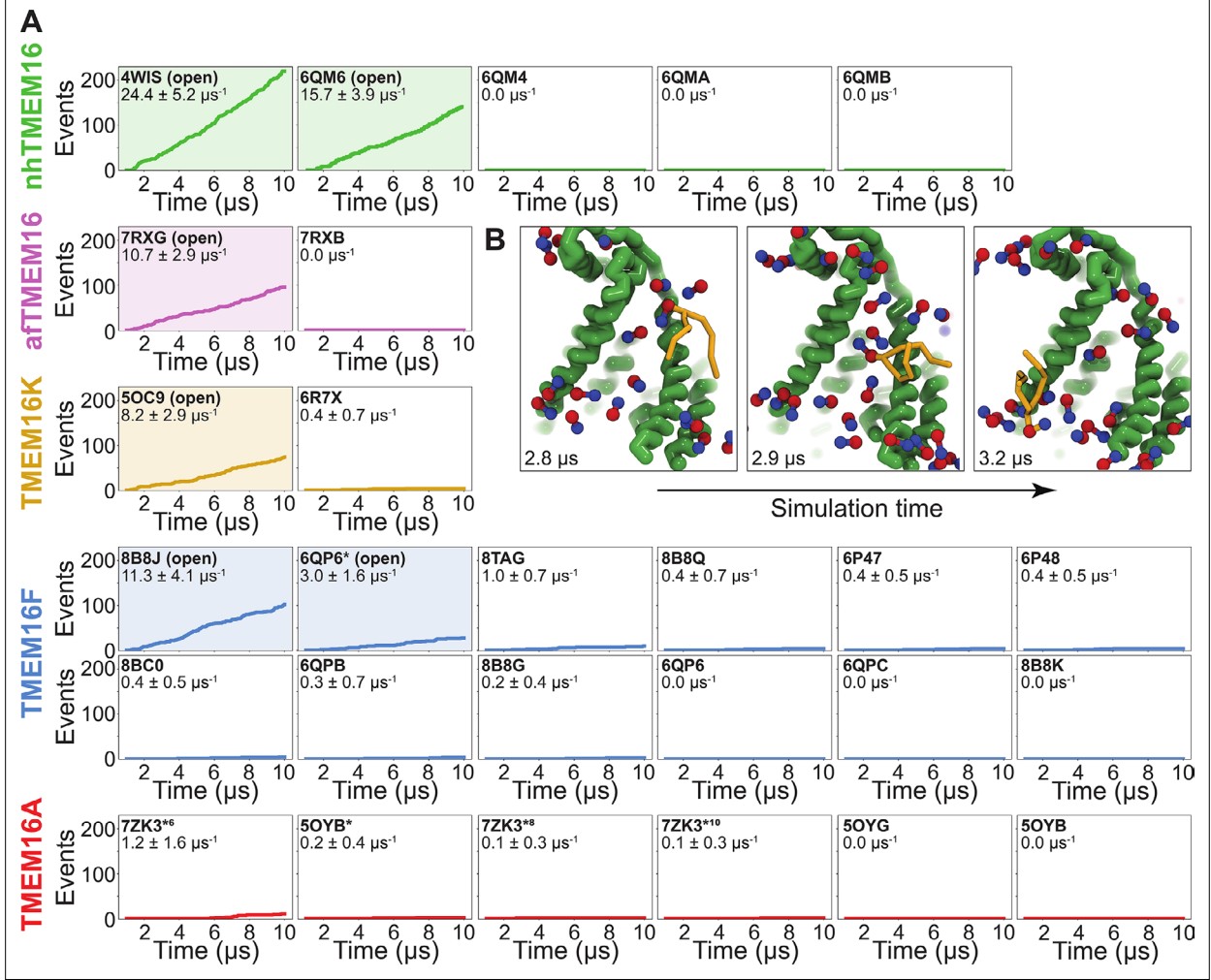

**Figure 2.** Simulated lipid scrambling differentiates closed/open conformations. (**A**) Accumulated scrambling events during coarse-grained molecular dynamics (CGMD) simulations of experimental and simulated (sim) structures of nhTMEM16 (green), afTMEM16 (violet), TMEM16K (gold), TMEM16F (blue), and TMEM16A (red). Inset values are the average rate and its standard deviation. Plots corresponding to structures described as 'open' in their original publications (PDB IDs 4WIS [*Brunner et al., 2014*], 6QM6 [*Kalienkova et al., 2019*], 7RXG [*Falzone et al., 2022*], 5OC9 [*Bushell et al., 2019*], 8B8J [*Arndt et al., 2022*], and 6QP6* [*Khelashvili et al., 2022*]) are shaded. (**B**) Snapshots of the open nhTMEM16 simulation (PDB ID 4WIS) showing a single scrambling event over time. The tail (yellow) of the scrambling lipid is explicitly shown, while all other lipids only show the phosphate (red)/choline (blue) headgroup.

The online version of this article includes the following figure supplement(s) for figure 2:

**Figure supplement 1.** Measuring lipid angles to detect scrambling events.

**Figure supplement 2.** Position traces of scrambling lipids in fungal TMEM16 simulations.

**Figure supplement 3.** Position traces of scrambling lipids in the open TMEM16K simulation.

**Figure supplement 4.** Position traces of scrambling lipids in TMEM16F simulations.

**Figure supplement 5.** Position traces of scrambling lipids in TMEM16A simulations.

observed little to no lipid scrambling in each case (*Figure 2A*). Others have shown that mutations at position F518 turn TMEM16F into a constitutively active scramblase (*Le et al., 2019b*). The F518H mutant (PDB ID 8B8J) is structurally characterized by a kink in TM3, and TM4 pulls away from TM6 35° compared to a closed WT TMEM16F structure (PDB ID 6QP6) (*Arndt et al., 2022*). In our simulations, TMEM16F F518H (PDB ID 8B8J) was the only system initiated directly from a solved structure that showed scrambling activity (11.3 ± 1.6 events per µs, *Figure 2—figure supplement 4A*). Additionally, we performed CGMD on a WT TMEM16F with a single open groove obtained from AAMD initiated from a closed-state structure (cluster 10 in *Khelashvili et al., 2022*, 6QP6* in *Figure 2A*).

**Table 1.** Number of scrambling events in and out of the canonical groove pathway.

Scrambling events where the lipid headgroup transitions between leaflets within 4.7 Å of the 1,2-dioleoyl-sn-glycero-3-phosphatidylcholine (DOPC) maximum density pathway. All other events were considered 'out-of-the-groove'. For the full list of simulations and scrambling rates, see *Source data 1*.

| Homolog | PDB code | # of in-the-groove events | # of out-of-the-groove events | Total | Average scrambling rate ($\mu s^{-1}$) |
|---|---|---|---|---|---|
| nhTMEM16 | 4WIS | 219 | 1 | 220 | 24.4 ± 5.2 |
| nhTMEM16 | 6QM6 | 141 | 0 | 141 | 15.7 ± 3.9 |
| afTMEM16 | 7RXG | 96 | 0 | 96 | 10.7 ± 2.9 |
| TMEM16K | 5OC9 | 66 | 8 | 74 | 8.2 ± 2.9 |
| TMEM16K | 6R7X | 0 | 4 | 4 | 0.4 ± 0.7 |
| TMEM16F F518H | 8B8J | 98 | 4 | 102 | 11.3 ± 4.1 |
| TMEM16F | 6QP6* | 24 | 3 | 27 | 3.0 ± 1.6 |
| TMEM16F T137Y | 8TAG | 0 | 9 | 9 | 1.0 ± 0.7 |
| TMEM16F | 6P47 | 1 | 3 | 4 | 0.4 ± 0.5 |
| TMEM16F | 6P48 | 0 | 4 | 4 | 0.4 ± 0.5 |
| TMEM16F F518H/Q623A | 8BC0 | 0 | 4 | 4 | 0.4 ± 0.5 |
| TMEM16F F518H | 8B8Q | 0 | 4 | 4 | 0.4 ± 0.7 |
| TMEM16F F518H | 8B8G | 2 | 0 | 2 | 0.2 ± 0.4 |
| TMEM16F | 6QPB | 0 | 3 | 3 | 0.3 ± 0.7 |
| TMEM16A | 7ZK3*[6] | 0 | 11 | 11 | 1.2±1.6 |
| TMEM16A | 5OYB* | 2 | 0 | 2 | 0.2 ± 0.4 |
| TMEM16A | 7ZK3*[10] | 0 | 1 | 1 | 0.1 ± 0.3 |
| TMEM16A | 7ZK3*[8] | 0 | 1 | 1 | 0.1 ± 0.3 |

We observed moderate lipid scrambling activity (3.0 ± 1.6 events per µs), most of which happened through the open groove (*Figure 2—figure supplement 4B, C*). Although the rates of scrambling are higher for the mutant than the open WT TMEM16F, there were no noticeable differences in how lipids enter the pathway or how long they take to transition (*Figure 4—figure supplement 3*).

Finally, we simulated six structures of mouse TMEM16A, which functions as an ion channel but lacks lipid scrambling activity (*Paulino et al., 2017*). As expected, both the Ca²⁺-bound (PDB ID 5OYB) and the Ca²⁺-free (PDB ID 5OYG) experimental structures failed to induce scrambling in the CGMD simulations, as did one alternative and two ion conduction-competent structures that were obtained from AAMD (see Appendix 1—Methods for details). However, a TMEM16A state with an open hydrophilic groove predicted by Jia and Chen (5OYB*, simulations initiated from PDB ID 5OYB; *Jia and Chen, 2021*) did scramble a single lipid through each groove in a manner nearly identical to the scramblases (*Figure 2—figure supplement 5A, E* and *Figure 3—video 4*).

## Groove dilation is the main determinant for scrambling activity

The relative impact of membrane thinning versus TM4/TM6 groove opening on the lipid scrambling rate has long been debated in the TMEM16 field. One of the primary open questions is whether membrane thinning is *sufficient* for scrambling when the groove is closed (*Feng et al., 2024b*). In our CGMD simulations, 92% of the observed scrambling events occur along TM4 and TM6 with headgroups embedded in the open hydrated groove, in line with the credit card model, which we refer to as 'in-the-groove' scrambling (*Table 1*). To visualize how groove openness and membrane thinning relate to these events, we plotted the minimum distance between residues on TM4 and TM6 against the minimal thickness near the groove in our average membrane surfaces (see Methods for details) and colored each data point by scrambling rate in the groove (*Figure 3A*).

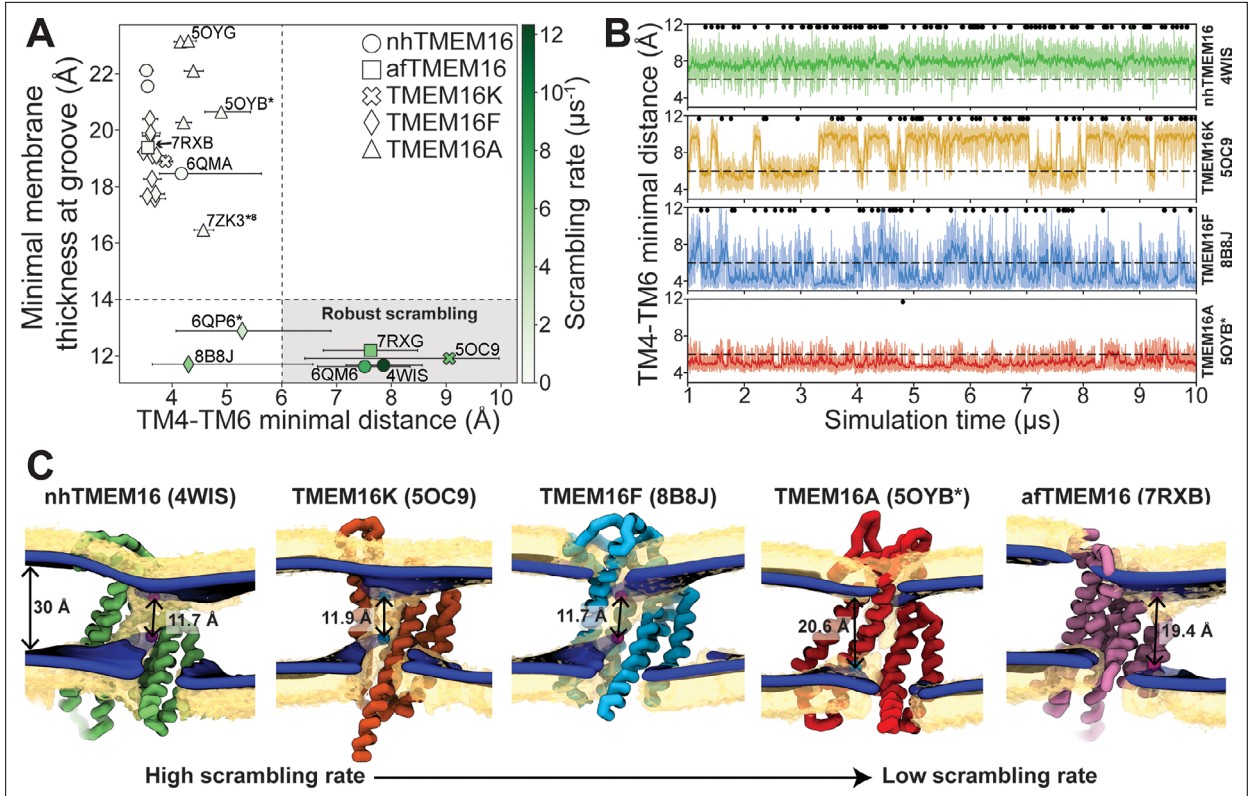

**Figure 3.** Lipid scrambling rates correlate with groove openness and membrane thinning. (**A**) The minimal membrane thickness at the groove plotted against the median width of the groove measured based on the minimal distance between any two residues on TM4 and TM6 of the groove with the most scrambling events. The lower and upper error bars represent the 25% (Q1) and 75% (Q3) quartiles, respectively. Each data point is colored by the scrambling rate through that same groove. Dashed lines define minimal TM4–TM6 distance and membrane thickness requirements for robust scrambling (shaded gray quadrant). (**B**) Simulation time traces of the TM4–TM6 minimal distance at the most scrambling-competent groove of 4WIS, 5OC9, 8B8J, and 5OYB* (top to bottom). The dashed line indicates the 6 Å threshold we defined for scrambling-competent groove opening. Black dots indicate time points at which a scrambling event is completed. The solid curve is a recursively exponentially weighted moving average with a smoothing factor 0.1, while the transparent curve is the raw distance values. (**C**) Density isosurfaces for 1,2-dioleoyl-sn-glycero-3-phosphatidylcholine (DOPC) headgroup beads (yellow) and average membrane surface calculated from the glycerol beads (blue) for representative nhTMEM16, TMEM16K, TMEM16F, TMEM16A, and afTMEM16 simulations. Panels are ordered left to right by decreasing scrambling rate. Cartoon beads and arrows in each image indicate the closest points between the inner and outer leaflet of the average surface.

The online version of this article includes the following video and figure supplement(s) for figure 3:

**Figure supplement 1.** Membrane deformations for simulated nhTMEM16 structures.

**Figure supplement 2.** Membrane deformations for simulated afTMEM16 structures.

**Figure supplement 3.** Membrane deformations for simulated TMEM16K structures.

**Figure supplement 4.** Membrane deformations for simulated TMEM16F structures.

**Figure supplement 5.** Membrane deformations for simulated TMEM16A structures.

**Figure supplement 6.** TM4 moves away from starting structure coordinates in open states.

**Figure 3—video 1.** Lipid scrambling by open $Ca^{2+}$-bound nhTMEM16 (PDB ID 4WIS).

https://elifesciences.org/articles/105111/figures#fig3video1

**Figure 3—video 2.** Lipid scrambling by open $Ca^{2+}$-bound TMEM16K chain B (PDB ID 5OC9).

https://elifesciences.org/articles/105111/figures#fig3video2

**Figure 3—video 3.** Lipid scrambling by open $Ca^{2+}$-bound TMEM16F F518H (PDB ID 8B8J).

https://elifesciences.org/articles/105111/figures#fig3video3

**Figure 3—video 4.** Lipid scrambling event 1/2 by simulated ion-conductive $Ca^{2+}$-bound TMEM16A (5OYB*).

https://elifesciences.org/articles/105111/figures#fig3video4

**Figure 3—video 5.** Lipid scrambling by simulated open $Ca^{2+}$-bound TMEM16F (PDB ID 6QP6*).

https://elifesciences.org/articles/105111/figures#fig3video5

*Figure 3 continued on next page*

*Figure 3 continued*

**Figure 3—video 6.** Lipid scrambling by open Ca²⁺-bound TMEM16K chain A (PDB ID 5OC9).
https://elifesciences.org/articles/105111/figures#fig3video6

Interestingly, *all* the TMEM16 structures included in this study thin the membrane to 23 Å or less, which is at least 7 Å thinner than the bulk membrane thickness (30 Å), regardless of scrambling activity (*Figure 3A*; *Figure 3—figure supplements 1–5*). We observed negligible scrambling activity (0–1 events in the groove) in grooves that fail to thin the membrane to less than 14 Å and at the same time do not or very rarely sample TM4-TM6 distances above 6 Å (*Figure 3A*, upper left quadrant). On the other hand, all active scramblers have a minimal bilayer thickness below 14 Å. Among these structures, we observed the highest scrambling rates in grooves that remain open, with TM4–TM6 distances above 6 Å, throughout most of the simulation (shaded region). To the left of this shaded area, there are two TMEM16F structures (PDB ID 8B8J and 6QP6*) that spent less than half of their simulation time in an open configuration (note large error bars and *Figure 3B*) and had scrambling rates similar to (PDB ID 8B8J) or less than half of (6QP6*) rates for the open scramblases (PDB IDs 6QM6, 7RXG, and 5OC9). Although these results indicate that scrambling rates are generally higher with thinner membranes and wider grooves, we want to clarify that lipids flowing into the upper and lower vestibules of the dilated grooves heavily contribute to the observed <14 Å membrane thickness (*Figure 3C*). Therefore, we argue that the extremely thin membranes are likely correlated with groove opening, rather than being an independent contributing factor to lipid scrambling. Thus, the major determinant of lipid scrambling by TMEM16s is dilation of the TM4/TM6 groove.

Upon closer inspection of TMEM16F, we noticed that hydrophobic residues (H/F518, W619, and M522) at the midpoint of the pathway, previously identified as an activation gate (*Le et al., 2019b*), dynamically swing open to sporadically allow lipids through (*Figure 3—figure supplement 6C*; *Figure 3—videos 3 and 5*). Although the distribution of the groove distances is similar for both TMEM16F structures that exhibit scrambling (*Figure 3—figure supplement 6B*), the WT open structure (6QP6*) has half the single subunit scrambling rate. We observed similar fluctuations in both subunits of the open asymmetric TMEM16K (PDB ID 5OC9) which transiently constrict the lipid pathway at Y366/I370/T435/L436 (*Figure 2—figure supplement 3A*, *Figure 3—figure supplement 6A, D*; *Figure 3—videos 2 and 6*). Again, we observed that the subunit with more scrambling activity (eight times more) spent more time in an open-groove configuration (*Figure 3—figure supplement 6B*). Time traces of the TM4–TM6 distances emphasize the two-state, discrete nature of the TMEM16K groove as it opens and closes, the consistently open nature of nhTMEM16 with small fluctuations, and then the frequent fluctuations of the TMEM16F F518H mutant, which has a running average that constantly flickers from 4 Å to the 7–8 Å range (*Figure 3B*). Qualitatively, scrambling occurs more frequently when the groove is open for TMEM16K and F (black dots in *Figure 3B*), while the consistently open Ca²⁺-bound nhTMEM16 structure (PDB ID 4WIS) allows lipid headgroups to scramble in an uninterrupted fashion (*Figure 3B*, *Figure 3—video 1*).

Although all structures of TMEM16A, which is not a scramblase, have negligible scrambling in the groove, we did observe two events for a predicted ion-conductive state (5OYB*) which samples an average TM4–TM6 distance very close, but just below, to the empirically determined 6 Å threshold for scrambling (*Figure 3B*). We observed lipid headgroups throughout the pathway, but just as for the TMEM16F and K structures, the flow of lipid is obstructed by residues at the center of the groove (I550, I551, and K645), and in TMEM16A they more rarely separate to allow lipids to pass (*Figure 3—figure supplement 6B*). Lipids are also notably more stagnant in the pore than in the open TMEM16Fs and appear to be stabilized by electrostatic interactions with two charged residues, E633 and K645 (*Figure 3—video 4*).

Despite these individual differences in groove dynamics, scrambling occurs in an identical manner across the family. Scrambling lipids move through the TM4/TM6 groove quickly, with dwell times for individual lipids below 20 ns. However, we observed longer dwell times for TMEM16K and TMEM16F at the groove constriction points, whereas in other scramblases, the dwell times are more evenly distributed along the groove (*Figure 4—figure supplements 3 and 4*). Among the scramblases, the free energy profile for lipids moving through the open groove is barrierless (<1 kT) (*Appendix 1—figure 3A*) with similar kinetics among the homologs and a mean diffusion coefficient between 10 and 16 Å²/ns (*Appendix 1—figure 4*). Scrambling events also enter and leave the groove at random

locations (*Figure 2—figure supplements 2–5*) with only 3–10% of events passing through the high-density lipid regions on lower TM4 and upper TM6/TM8 (*Figure 1B*; *Figure 4—figure supplement 2*). We previously identified four residues (E313, R432, K353, and E352) at the intracellular and extracellular entrances of the nhTMEM16 groove that we hypothesized help organize or stabilize scrambling lipids (*Bethel and Grabe, 2016*; *Figure 3—figure supplement 1A, B*). However, our CGMD of the same nhTMEM16 structure shows that although these residues have elevated contact frequencies, more than half of the contacts are made with bulk lipids that never scramble (*Figure 4—figure supplement 1C, D*). Lastly, the in-the-groove scrambling events were Poisson distributed for all open and transiently open scramblases (*Appendix 1—figure 5*), indicating lipids do not scramble in a regular or kinetically coordinated fashion.

## Water and ion content in the groove

To quantify how hydration of the groove or pore relates to scrambling, we measured the number of water permeation events along the pathway of maximum water density at the grooves (*Appendix 1—figure 2A*; all values in *Source data 1*; see Appendix 1—Methods for details). As expected, permeation through the closed scramblase structures was low, <30 events per μs on average, while dilated TM4/TM6 grooves (five out of six $Ca^{2+}$-bound) support 300–550 permeation events per μs on average. Nonetheless, even when the groove is inaccessible to lipids in closed and intermediate states, including the TMEM16A ion channel path, it remains hydrated with the waters shielded from the hydrophobic core of the membrane (*Appendix 1—figure 2A*, *closed*). We qualitatively observed that in open grooves, the water is exposed to the membrane core and lipid headgroups occupy the fully hydrated groove to bridge the leaflets (*Appendix 1—figure 2A*, *open*) as seen in fully atomistic simulations (*Bushell et al., 2019*; *Stansfeld et al., 2015*; *Bethel and Grabe, 2016*; *Jiang et al., 2017*; *Khelashvili et al., 2019*; *Lee et al., 2018*; *Cheng et al., 2022*; *Kostritskii and Machtens, 2021*; *Le et al., 2019b*; *Khelashvili et al., 2022*; *Jia et al., 2022*).

We also observed spontaneous permeation of $Na^+$ and $Cl^-$ ions through the scramblase TMEM16 grooves and TMEM16A pore (*Appendix 1—figure 2B*; number of permeation events in *Source data 1*), in line with the known ion-conducting capacity of these proteins (*Yang et al., 2012*; *Martins et al., 2011*; *Suzuki et al., 2010*; *Malvezzi et al., 2013*; *Bushell et al., 2019*; *Lee et al., 2016*; *Scudieri et al., 2015*; *Falzone et al., 2019*; *Cheng et al., 2022*; *Kostritskii and Machtens, 2021*). Of the fungal structures, only the scrambling-competent open states sampled multiple ion permeation events with $Ca^{2+}$-bound nhTMEM16 (PDB ID 4WIS) showing highest conductance followed by $Ca^{2+}$-free nhTMEM16 (PDB ID 6QM6), which was 3 times lower, and then $Ca^{2+}$-bound afTMEM16 (PDB ID 7RXG), which was another three times lower again. We also measured cation-to-anion selectivity ratios of 5.1, 3.2, and 6 for each simulation, respectively, computed from the ratio of total counts ($P_{Na}/P_{Cl}$). Our simulations are consistent with experiments showing that both fungal scramblases transport anions and cations (*Lee et al., 2016*), and both are weakly cation selective ($P_K/P_{Cl}$ = 1.5 for afTMEM16 based on experiment [*Malvezzi et al., 2013*] and $P_{Na}/P_{Cl}$ = 8.7 for nhTMEM16 based on AAMD [*Kostritskii and Machtens, 2021*]). Our CGMD simulations also sample ion conduction through open $Ca^{2+}$-bound TMEM16F F518H (PDB ID 8B8J, $P_{Na}/P_{Cl}$ = 1.3), which had the most ion permeation events (*Hunter, 2007*) across the family, simulated open TMEM16F (6QP6*, $P_{Na}/P_{Cl}$ = 0.33), and open TMEM16K (PDB ID 5OC9, $P_{Na}/P_{Cl}$ = 1.8). This latter result on TMEM16K qualitatively agrees with experiment showing a slight cation preference (*Bushell et al., 2019*), while experimental results for TMEM16F are more complex as its ion selectivity depends on membrane potential and divalent/monovalent cation concentrations (*Nguyen et al., 2021*; *Stabilini et al., 2021*; *Ye et al., 2019*). Our simulation of the TMEM16F F518H mutant in 150 mM NaCl is most close to whole cell recordings performed in intracellular 150 mM NaCl and 15 μM $Ca^{2+}$ where $P_{Na}/P_{Cl}$ = 1.0 ± 0.1 (*Ye et al., 2019*), which is very similar to our simulated value of 1.3. With regard to the selectivity values reported here, it is important to note that we observed less than 20 total events each for WT TMEM16F (6QP6*), afTMEM16, and TMEM16K (see *Source data 1*), and therefore, the values are prone to statistical error. We are more confident in the ratios reported for TMEM16F F518H and nhTMEM16 (PDB ID 4WIS) as those emitted 99 and 61 events, respectively.

Finally, TMEM16A (7ZK3*[8]) had four $Cl^-$ and no $Na^+$ permeation events, consistent with its experimentally measured anion selectivity ($P_{Na}/P_{Cl}$ = 0.1; *Peters et al., 2018*). Interestingly, we did not observe $Cl^-$ permeation in any of the other computationally predicted TMEM16A structures (5OYB*,

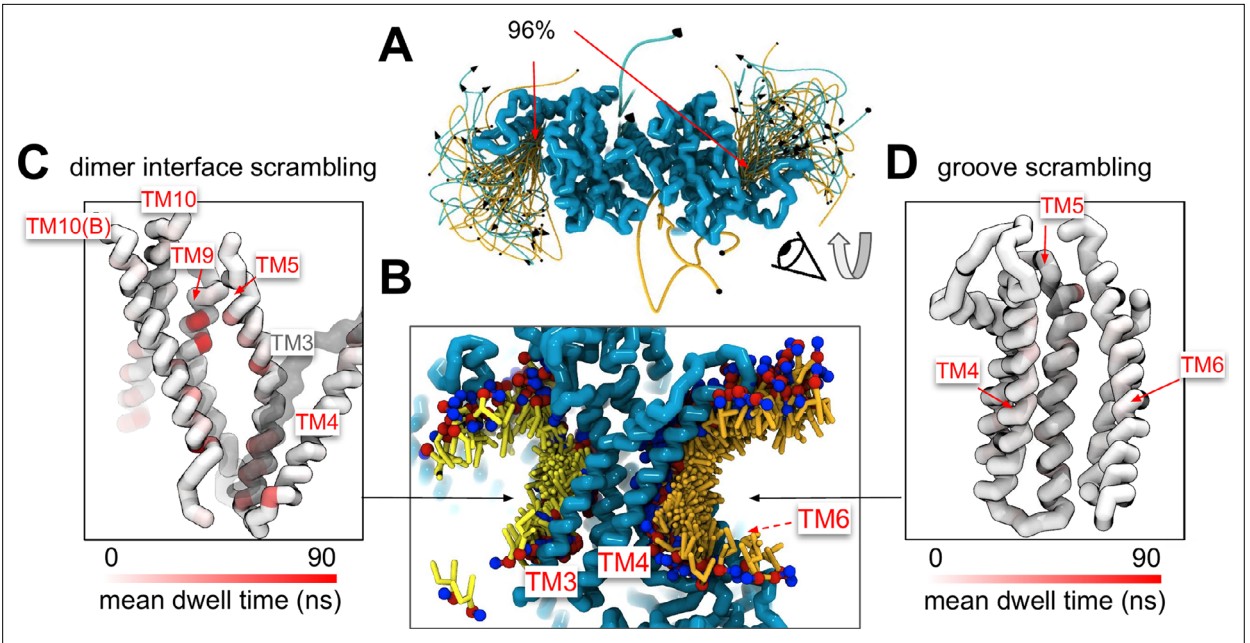

**Figure 4.** Lipid scrambling events and lipid–protein residue contact in the dimer interface and canonical TM4/TM6 groove. (**A**) Traces of all scrambling lipids in a TMEM16F (PDB ID 8B8J) simulation. Lipid scrambling from the inner to outer leaflet is illustrated as cyan traces and from the outer to inner leaflet as yellow traces. (**B**) Cartoon depiction of two individual inward scrambling events along the TM4/TM6 groove (orange tail with red/blue headgroup) and the dimer interface (yellow tail with red/blue headgroup) with multiple snapshots over time. Only the headgroup, first and second tail beads are shown for clarity. Protein backbone colored by mean lipid headgroup interaction (dwell) time at the TMEM16F dimer interface (**C**) and TM4/TM6 groove (**D**).

The online version of this article includes the following figure supplement(s) for figure 4:

**Figure supplement 1.** Contact analysis of lipid headgroup high-density sites identified from previous AA simulation.

**Figure supplement 2.** Position traces of scrambling lipids with total lipid headgroup density.

**Figure supplement 3.** Average duration of interaction between scrambling lipids and TM4/TM6 groove lining residues.

**Figure supplement 4.** Dwell time distribution and contact frequency for TM4/TM6 groove lining residue across homologs.

**Figure supplement 5.** Lipids enter the dimer interface in atomistic and CG simulations of TMEM16F.

**Figure supplement 6.** Dimer interface hydrophobicity and lipid positions.

**Figure supplement 7.** Dwell time analysis for scrambling events observed at TMEM16F F518H mutant dimer interface.

7ZK3*[8], and 7ZK3*[10]), while AAMD simulations of these structures all reported Cl⁻ conduction (*Jia and Chen, 2021*).

## Scrambling also occurs out-of-the-groove

A minority of our observed scrambling events (8%) occurred outside of the hydrophilic groove between TM4 and TM6. Surprisingly, most of these events happened at the dimer interface with lipids inserting their headgroups into the cavity outlined by TM3 and TM10 (*Figure 4A, B*; *Figure 2—figure supplements 2–5*). We only observed scrambling at this location in simulations of the mammalian homologs. In atomistic simulations of a closed $Ca^{2+}$-bound TMEM16F (PDB ID 6QP6), we observed a similar flipping event for a POPC lipid into the dimer interface (*Figure 4—figure supplement 5*). Although the dimer interface is largely hydrophobic, there are a few polar and charged residues in the cavity near the membrane core, and water is present in the lower half of the cavity (*Figure 4—figure supplement 6*). In fact, the headgroup of the lipid in our atomistic simulation of TMEM16F interacts with a glutamate (E843) and lysine (K850) on TM10 near the membrane midplane (*Figure 4—figure supplement 5*). Lipids that scramble at the dimer interface interact with the protein up to 10-fold longer on average than those in the canonical groove (*Figure 4C, D*). The most prolonged interactions occur at sites containing aromatic residues into which the lipid tails intercalate (*Figure 4—figure supplement 7*).

There were five more out-of-the-groove events, including one that occurred across a closed TM4/TM6 groove of $Ca^{2+}$-bound TMEM16F (PDB ID 6P47). From all our observed scrambling events, this is the only one that fits the postulated out-of-the-groove definition where scrambling is expected to take place near TM4/TM6 but without inserting into the groove (*Feng et al., 2024b*; *Appendix 1—figure 6A*). Two events occurred concurrently along TM6 and TM8 again near the hydrophilic groove of a $Ca^{2+}$-bound closed TMEM16F (PDB ID 8TAG; *Appendix 1—figure 6A, B*). Lastly, two events occurred along TM3 and TM4, one near the canonical TM4/TM6 groove of an open nhTMEM16 (PDB ID 4WIS) and the other adjacent to the pore of an ion-conductive TMEM16A (7ZK3[*8]; *Appendix 1—figure 6C, D*). In each of these five out-of-the-groove events, the scrambling lipid traverses with two to four water molecules around its headgroup.

## Discussion

Previous all-atom simulations of TMEM16 have captured partial translocations or – at most – a handful of complete scrambling events (e.g., *Bethel and Grabe, 2016*; *Lee et al., 2018*; *Khelashvili et al., 2022*) due to the challenges inherent in simulating molecular events on the low microseconds time scale. Although this small number of AAMD-derived scrambling events yielded key insights into specific protein–lipid interactions and scrambling pathways, they cannot provide rigorous statistics on scrambling rates, nor can they be leveraged to perform a large high-throughput comparison between the various family members. To circumvent sampling issues, we used CGMD to systematically quantify lipid scrambling by five TMEM16 family members and relate their scrambling competence to their structural characteristics and ability to distort the membrane. Our simulations correctly differentiate between open and closed conformations across the five family members, consistent with a recent study that showed good qualitative agreement between in vitro and in silico lipid scrambling using the same Martini 3 force field on a diverse set of proteins, including some TMEM16s (*Tsuji et al., 2019*). In addition to lipid scrambling ability, our results are in accord with the general finding that TMEM16s show very little to no ion selectivity, although permeability ratios vary depending on ion concentrations and lipid environments (*Kostritskii and Machtens, 2021*; *Nguyen et al., 2021*; *Stabilini et al., 2021*; *Ye et al., 2019*). Because the simulation conditions and system setups were identical in all our simulations, we are in a unique position to directly compare a host of biophysical properties between different TMEM16 family members and their structures to answer ongoing questions in the field.

In our simulations, *all* TMEM16 structures thin the membrane by at least 7 Å, while some pinch the membrane by as much as 18 Å resulting in leaflet-to-leaflet distances at the groove of just 12 Å (*Figure 3A, D*). We (*Bethel and Grabe, 2016*) and others *Bushell et al., 2019*; *Khelashvili et al., 2019*; *Falzone et al., 2022*; *Arndt et al., 2022*; *Feng et al., 2019* have hypothesized that thinning lowers the physical and energetic barrier for lipid scrambling, but what is surprising is that even non-scrambling, closed-groove structures elicit such large membrane distortions. For instance, several of the closed-groove TMEM16F structures and the ion channel TMEM16A (7ZK3[*8]) compress the membrane 13–14 Å. Despite this large deformation, these conformations do not induce scrambling. On the other hand, structures with dilated grooves exhibit robust scrambling and thin the membrane another 3–4 Å, resulting in the most distorted bilayers. However, because this extreme membrane thinning is coupled to lipid entry into the upper and lower vestibules upon groove opening, it is difficult to determine how much the membrane thinning alone contributes to the resulting scramblase activity. Thus, we conclude that groove dilation is the ultimate trigger for rapid lipid scrambling, and the importance of membrane thinning to modulating scrambling rates has yet to be determined.

Of the scrambling-competent TMEM16 structures, the open-groove nhTMEM16 (PDB ID 4WIS) is the fastest scrambler, with a rate twice as high as the other homologs (*Figures 2A and 3A, B*). Yet on average, its groove width and membrane thinning are similar (within 1–2 Å) to the other robust scramblers nhTMEM16 (PDB ID 6QM6), afTMEM16 (PDB ID 7RXG), and TMEM16K (PDB ID 5OC9) (*Figure 3A*). This suggests that there are other features that impact the rates, for example, the shape of the membrane distortion, groove dynamics, and residues lining the groove. Another feature we have not explored is mixed membranes and membranes of shorter or longer chain length, which we expect would alter lipid scrambling rates. For example, TMEM16K resides in the ER membrane which is thinner than the plasma membrane (*Bushell et al., 2019*; *Petkovic et al., 2019*; *Bruininks et al., 2019*), and TMEM16K scrambling rates increase 10-fold in thinner membranes (*Bushell et al., 2019*).

Experimental scrambling assays performed by different groups have reported basal level scramblase activity in the absence of $Ca^{2+}$ for fungal and mammalian dual-function scramblases (*Malvezzi et al., 2013*; *Bushell et al., 2019*; *Lee et al., 2016*; *Brunner et al., 2014*; *Lee et al., 2018*; *Malvezzi et al., 2018*; *Watanabe et al., 2018*). It is unknown where closed-groove scrambling takes place on the protein (*Feng et al., 2024a*) and simulations have never reported such events despite Li and co-workers reporting scrambling events for simulations initiated from closed TMEM16A, TMEM16K, and TMEM16F (*Tsuji et al., 2019*), which may have also been sampled in these trajectories. In aggregate, we observed 60 scrambling events that do not follow the credit card model and occur 'out-of-the-groove' (*Table 1*). Nearly all these events (56/60) happen at the dimer interface between TM3 and TM10 of the opposite subunit, hereon referred to as the dimer cleft. Curiously, we do not observe scrambling at this location for any of the fungal structures. Although mammalian TMEM16s have a ~4–5 Å wider gap on average at the lower leaflet dimer cleft entrance than the open fungal TMEM16s, we do not always observe scrambling at such distances and sometimes do not observe any scrambling when the cleft is at its widest (*Appendix 1—figure 7*). For all structures, we see lipids from both leaflets intercalate between TM3 and TM10 (*Figure 4—figure supplement 6*), which is consistent with lipid densities in cryo-EM nanodiscs images of fungal TMEM16s (*Falzone et al., 2022*; *Feng et al., 2024a*) and TMEM16F (*Feng et al., 2019*). Based on our simulations, this interface may be a source for $Ca^{2+}$-independent scrambling.

It is unclear whether the out-of-the-groove events we have observed reflect the same closed-groove scrambling activity seen in experimental assays (*Lee et al., 2018*; *Malvezzi et al., 2018*; *Falzone et al., 2022*; *Feng et al., 2024a*). Also, it is possible that we have missed slow or rare out-of-the-groove events due to limited sampling. One way to assess these points is to ask whether the relative scrambling rates observed in ±$Ca^{2+}$ are similar to the relative rates from our simulation with open/closed hydrophilic grooves. Feng et al. reported a 7- to 18-fold increase in scrambling rate by nhTMEM16 in the presence of $Ca^{2+}$ compared to $Ca^{2+}$-free conditions (*Feng et al., 2024a*). Based on our open-groove count of 220, we would expect 12–30 events for the closed-groove states, but we observed no events. However, Watanabe et al. reported a six- to seven-fold increase in scrambling rate by TMEM16F in the presence of $Ca^{2+}$ compared to $Ca^{2+}$-free conditions (*Watanabe et al., 2018*), which is consistent with the seven- to nine-fold increase revealed in our simulations between closed $Ca^{2+}$-free TMEM16F structures (PBD IDs 6P47 and 6QPB) and the WT open $Ca^{2+}$-bound TMEM16F (6QP6*). It is possible that out-of-the-groove scrambling is highly dependent on the membrane composition, as discussed earlier, and the scrambling ratios we observe in DOPC may be different from the experimental rates determined in different lipids. This cannot be addressed without additional studies. That said, we are encouraged by the high-level correspondence in TMEM16F – we observe much higher scrambling rates through the open grooves and much smaller flipping rates elsewhere on the protein or with closed-groove structures, suggesting that our simulations may be revealing aspects of $Ca^{2+}$-independent scrambling in mammalian family members.

With regard to predicting absolute rates, our simulations correctly distinguish scrambling-competent structures from non-competent scramblers, but direct comparison of our rates with experimental values (that tend to be 2–3 orders of magnitude slower) should be interpreted qualitatively. For example, single-molecule analysis yielded a scrambling rate of 0.04 events per µs for TMEM16F (*Watanabe et al., 2018*), whereas we find 3 and 11.3 events per µs for our scrambling-competent TMEM16F structures 6QP6* and PDB ID 8B8J, respectively. Malvezzi et al. estimated a similar scrambling rate of 0.02 events per µs for afTMEM16 using a liposome-based assay, while we find 10.7 events per µs (*Malvezzi et al., 2018*). We will highlight three potential explanations for such discrepancy. First, it is well established that the Martini model increases diffusion dynamics by a factor ~4 due to the lower friction between CG beads and reduced configurational entropy compared to more chemically detailed representations (*Bruininks et al., 2019*). Second, the energy barrier for a PC headgroup to traverse the DOPC bilayer in the absence of protein is reduced in Martini 3 compared to Martini 2 and AAMD (*Bartoš et al., 2024*). It is not trivial to predict how this reduction affects protein-mediated lipid scrambling, but it is likely to increase observed flipping rates compared to the more realistic AAMD. Third, as shown in *Figure 3B*, the Martini 3 elastic network used to restrain the protein backbone in our simulations allows a small degree of flexibility during simulations, which may increase scrambling. For instance, the groove of the open nhTMEM16 structure 4WIS enlarges by ~3 Å during our Martini 3 simulations compared to the starting experimental structure and our Martini 2 simulations, and

this dilation correlates with greater scrambling (*Appendix 1—figure 8A, B*). We also analyzed previously published CHARMM36 AAMD trajectories starting from the same structure (*Bethel and Grabe, 2016*) and observed that while these simulations do show some degree of dilation, as we observe with Martini 3, they generally stay closer to the experimental structure (*Appendix 1—figure 8B*). In addition to the open nhTMEM16 structure, we observed similar subtle movements in the TM4 helix for open $Ca^{2+}$-bound structures of afTMEM16, TMEM16K, TMEM16F, and TMEM16A that appear to enlarge the TM4/TM6 outer vestibule (*Figure 3—figure supplement 6A*). Others have reported that AAMD simulations sample spontaneous dilation of the groove/pore to confer either scramblase activity for WT (*Le et al., 2019b*; *Khelashvili et al., 2022*) and mutant (*Jia et al., 2022*) TMEM16F or ion channel activity for TMEM16A structures (*Le et al., 2019a*; *Yu et al., 2019*; *Tembo et al., 2019*). These movements away from the experimentally solved structures may be due to the inaccuracy of our atomistic and CG force fields or differences in the model and experimental membrane/detergent environments, but more work is needed to assess whether these dilations reflect physiologically relevant conformational states. CG simulations of closed-groove structures lack such dilations, because the backbones of TM4 and TM6 are in close enough proximity (<10 Å) to be connected by the elastic network that the Martini model requires to maintain proper secondary and tertiary structure (e.g., PDB ID 6QM4, see *Appendix 1—figure 8C, D*). The recent GōMartini 3 model replaces the harmonic bonds of the elastic network with Lennard–Jones potentials that vanish as residues separate, potentially making this an excellent model for sampling groove opening and closing (*Souza et al., 2024*; *Appendix 1—figure 8E*).

Finally, we end by discussing the observed in- and out-of-the-groove scrambling for the putative ion-conducting states of TMEM16A. The low number of recorded events (11 for the highest and 1 for the lowest) may be consistent with the lack of experimentally measured scramblase activity (*Han et al., 2019*; *Gyobu et al., 2017*), for the reasons discussed in the last paragraph. Consistent with our low computational rate, we also computed an energy barrier for lipid movement through the TMEM16A groove 5.5-fold higher than the scramblase barriers (*Appendix 1—figure 3B*). In simulations of our three predicted ion-conductive states of TMEM16A (7ZK3*[6,8,10]) lipid headgroups insert into the lower and upper vestibule of the pore. Compared to the inhibitor-bound structure (PDB ID 7ZK3), the outer vestibule of these conductive states is notably more dilated. We observe four $Cl^-$ permeation events by 7ZK3*[8] through the partially lipid-lined groove. Surprisingly, our simulation of the predicted TMEM16A conductive state from Jia and Chen did at times feature a fully lipid-lined groove, similar to the proteolipidic pore found in dual-function members (*Jia and Chen, 2021*; *Figure 1D*, *Figure 3—video 1*); however, we did not observe any ion permeation events from this configuration, which may be a consequence of the configuration not being physiologically relevant, the Martini 3 force field not being ideal for $Cl^-$/lipid/protein interactions, or something else. It is intriguing that while TMEM16A has lost experimentally discernible scrambling activity, it still deforms and thins the membrane (*Figure 3A*). Coupled with our observation that groove widening allows lipids to enter, we wonder if it retains thinning capabilities to facilitate partial lipid insertion to promote $Cl^-$ permeation. This hypothesis has been stated before (*Whitlock and Hartzell, 2016*), and structural evidence for this proteolipidic ion channel pore has recently been reported for the OSCA1.2 mechanosensitive ion channel, which adopts the TMEM16 fold, yet it does not scramble lipids (*Lowry et al., 2024*; *Han et al., 2024*).

## Materials and methods
### Coarse-grained system preparation and simulation details
For each simulated structure, missing loops with less than 16 residues were modeled using the loop building and refinement procedures MODELLER (version 10.2, *Sali and Blundell, 1993*). Further details on which loops were included are in *Appendix 1—table 1*. For each stretch of *N* missing residues, 10 × *N* models were generated. We then manually assessed the 10 lowest DOPE scoring predictions and selected the best model based on visual inspection. Models were inserted symmetrically into the original experimental dimer structure except for PDB IDs 8BC0, 8TAG, and 5OC9 which were published as asymmetric structures.

Setup of the CG simulation systems was automated in a python wrapper script adapted from MemProtMD (*Stansfeld et al., 2015*). After preparing the atomistic structure using pdb2pqr (*Jurrus*

*et al., 2018*), the script predicted protein orientation with respect to a membrane with memembed (*Nugent and Jones, 2013*). Then, martinize2 (*Kroon et al., 2022*) was employed to build a Martini 3 CG protein model. Secondary structure elements were predicted by DSSP (*Kabsch and Sander, 1983*) and their inter- and intra-orientations within a 5–10 Å distance were constrained by an elastic network with a 500 kJ mol$^{-1}$ nm$^{-2}$ force constant (unless specified otherwise). CG Ca$^{2+}$ ions (bead type 'SD' in Martini 3) were inserted at their respective positions based on the original protein structure and connected to coordinating (≤6 Å) Asp and/or Glu side chains by a harmonic bond with a 100 kJ mol$^{-1}$ nm$^{-2}$ force constant. A DOPC membrane was built around the CG protein structure using *insane* (*Wassenaar et al., 2015*) in a solvated box of 220 × 220 × 180 Å$^3$, with 150 mM NaCl. Systems were charge-neutralized by adding Cl$^-$ or Na$^+$ ions. For each system, energy minimization and a 2-ns NPT equilibration were performed. All systems were simulated for 10 μs in the product, ion phase and the first microsecond was excluded from all analyses for equilibration.

All CGMD simulations were performed with Gromacs (version 2020.6; *Abraham et al., 2015*) and the Martini 3 force field (version 3.0.0). A 20 fs time step was used. Reaction-field electrostatics and Van der Waals potentials were cut off at 1.1 nm (*de Jong et al., 2016*). As recommended by *Kim et al., 2023*, the neighbor list was updated every 20 steps using the Verlet scheme with a 1.35 nm cut-off distance. Temperature was kept at 310 K using the velocity rescaling (*Bussi et al., 2007*) thermostat ($\tau_T$ = 1 ps). The pressure of the system was semi-isotropically coupled to a 1-bar reference pressure by the Parrinello–Rahman (*Parrinello and Rahman, 1981*) barostat ($\tau_P$ = 12 ps, compressibility = 3 × 10$^{-4}$).

## Lipid headgroup and water density calculations

First, each protein subunit was individually aligned in *x*, *y*, and *z* to their starting coordinates. Atomistic simulations were filtered for trajectory frames with T333–Y439 Cα distance >15 Å giving a total of ~2085 ns of aggregate simulation time. Then the positions of all PC headgroup beads were tracked over time and binned in a 100 × 100 × 150 Å grid with 0.5 Å spacing centered on two residues near the membrane midplane on TM4 and TM6 using a custom script that includes MDAnalysis methods (*Gowers et al., 2016*; *Michaud Agrawal et al., 2011*). Density for water beads was calculated in the same way. Density in each cell was then averaged from each chain and for atomistic simulations averaged from all eight independent simulations.

## Scrambling analysis

Lipid scrambling was analyzed as described by *Li et al., 2024*. For every simulation frame (1 ns$^{-1}$ sampling rate), the angle between each individual DOPC lipid and the *z*-axis was calculated using the average of the vectors between the choline (NC3) bead and the two last tail beads (C4A and C4B), see *Figure 2—figure supplement 1A*. We applied a 100 ns running average to denoise the angle traces. Lipids that reside in the upper leaflet are characterized by a 150° angle, and lipids in the lower leaflet have a 30° angle. Scrambling events were counted when a lipid from the upper leaflet passed the lower threshold at 35° or, vice versa, when a lipid from the lower leaflet passed the upper threshold at 145° (see *Figure 2—figure supplement 1B*). These settings are more stringent than the thresholds used by Li et al. (55° and 125°, respectively) to prevent falsely counted partial transitions (*Li et al., 2024*). A 1-μs block averaging was applied to obtain averages and standard deviations for the scrambling rates.

## Groove dilation analysis

The residues chosen for measuring the minimum distance between TM4 and TM6 were located within ~6 Å in *z* (1–2 α-helix turns) of the path node with the minimum net flux of water (see Appendix 1). The residues used for each homolog were as follows: 327–339 and 430–452 for nhTMEM16, 319–331 and 426–438 for afTMEM16, 365–377 and 434–446 for TMEM16K, 512–424 and 613–625 for TMEM16F, and 541–553 and 635–647 for TMEM16A. Distances were calculated using a custom script that includes MDAnalysis methods (*Gowers et al., 2016*; *Michaud Agrawal et al., 2011*).

## Quantification of membrane deformations

First, using Gromacs (gmx trjconv), MD trajectories were aligned in the *xy*-plane such that the longest principal axis defined by the initial positions of TM7 and TM8 aligned to the global *y*-axis. Average membrane surfaces were calculated from the aligned MD trajectories as outlined

previously (*Bethel and Grabe, 2016*) using a custom python script based on MDAnalysis *Michaud Agrawal et al., 2011* and SciPy (*Virtanen et al., 2020*). The positions of each lipid's glycerol beads (GL1 and GL2) were linearly interpolated to a rectilinear grid with 1 Å spacing. Averaging over all time frames (again, discarding the first 1 µs for equilibration) yielded a representative upper and lower leaflet surface. Grid points with a lipid occupancy below 2% were discarded. Clusters of grid points that were disconnected from the bulk membrane surface were discarded. The minimal membrane thickness was calculated as the minimal distance between any two points on the opposing ensemble-averaged surfaces (e.g., *Figure 3C*). Crucially, in the case of lipid scrambling simulations like the ones described here, lipids were assigned to the upper/lower leaflet separately for every time frame.

### Protein–lipid contact and dwell time analysis

Using the full 10 µs simulation where each protein subunit was individually aligned in $x$, $y$, and $z$, we analyzed protein–lipid interactions by measuring distances between the protein's outermost sidechain bead (except for glycine, which only has backbone bead) and the lipid's choline (NC3) or phosphate (PO4) bead for every nanosecond using custom scripts with Scipy methods (*Virtanen et al., 2020*). Contacts were defined as distances below 7 Å. Contact frequency was calculated as the fraction of simulation frames where a contact occurred, averaged over two monomers. Dwell time was measured as the duration of consecutive contacts, allowing breaks up to 6 ns to account for transient fluctuations of lipid configuration. For each residue, we selected either the choline or phosphate bead based on which yielded the higher average dwell time. To visualize the result, we used averaged dwell time of the top 50% longest dwelling events at each residue to generate a color-coded representation of the protein structure (*Figure 4C, D*; *Figure 4—figure supplement 3*).

### Simulation and data visualization

Each simulation video and all simulation snapshots with lipid headgroup coordinate densities and traces, average membrane surfaces, and protein colored by lipid contact/dwell time were rendered using VMD (*Humphrey et al., 1996*). Images of TMEM16A atomistic starting structures were rendered using ChimeraX (*Pettersen et al., 2021*). All plots were generated using the Matplotlib graphics package (*Hunter, 2007*).

## Acknowledgements

Neville Bethel performed the all-atom simulations of nhTMEM16 reported in *Bethel and Grabe, 2016*. Paola Bisignano performed the loop modeling for many of the TMEM16F structures. George Khelashvili generously shared the simulated open-groove structure of TMEM16F from *Khelashvili et al., 2022*. We thank Wenlei Ye for his helpful comments on our manuscript. We thank Andrew Natale and Yessica Gomez, who helped develop the membrane deformation analysis scripts.

## Additional information

### Funding

| Funder | Grant reference number | Author |
| --- | --- | --- |
| National Science Foundation | Graduate Research Fellowship Program 2038436 | Christina Alexandra Stephens |
| National Institutes of Health | R01 GM137109 | Michael Grabe |
| University of California, San Francisco | Discover Fellowship Program | Christina Alexandra Stephens |

The funders had no role in study design, data collection, and interpretation, or the decision to submit the work for publication.

## Author contributions
Christina Alexandra Stephens, Conceptualization, Data curation, Formal analysis, Funding acquisition, Visualization, Methodology, Writing – original draft, Writing – review and editing; Niek van Hilten, Lisa Zheng, Conceptualization, Data curation, Formal analysis, Visualization, Methodology, Writing – original draft, Writing – review and editing; Michael Grabe, Conceptualization, Resources, Supervision, Funding acquisition, Writing – original draft, Project administration, Writing – review and editing

## Author ORCIDs
Christina Alexandra Stephens ⓘ https://orcid.org/0000-0001-9382-003X
Niek van Hilten ⓘ https://orcid.org/0000-0003-1204-2489
Lisa Zheng ⓘ https://orcid.org/0000-0002-8640-2443
Michael Grabe ⓘ https://orcid.org/0000-0003-3509-5997

Reviewer #1 (Public review): https://doi.org/10.7554/eLife.105111.3.sa1
Reviewer #2 (Public review): https://doi.org/10.7554/eLife.105111.3.sa2
Reviewer #3 (Public review): https://doi.org/10.7554/eLife.105111.3.sa3
Author response https://doi.org/10.7554/eLife.105111.3.sa4

# Additional files

## Supplementary files
MDAR checklist

Source data 1. Raw simulation data for plots in *Figures 2 and 3*, *Appendix 1—figure 7*, and ion selectivity values provided in the main text.

## Data availability
All code and files used to generate MD trajectories and scripts to generate main figures and analyze MD trajectories are available on Zenodo: https://zenodo.org/records/15839331. Source data files have been provided for averages of data plotted in Figures 2 and 3, and Appendix 1—Figure 2.

The following dataset was generated:

| Author(s) | Year | Dataset title | Dataset URL | Database and Identifier |
| --- | --- | --- | --- | --- |
| Stephens CA, Hilten van N, Zheng L, Grabe M | 2025 | Simulation-based survey of TMEM16 family reveals that robust lipid scrambling requires an open groove | https://doi.org/10.5281/zenodo.15839331 | Zenodo, 10.5281/zenodo.15839331 |

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

## Appendix 1

### Methods

#### Additional system preparation and simulation details

##### Starting structure selection for coarse-grained (CG) simulations

We simulated 23 out of the 62 cryo-EM and X-ray TMEM16 homodimer structures available at the time (*Appendix 1—table 1*). We chose not to include structures with greater than 4 Å global resolution (apart from TMEM16A PDB ID 5OYG which is the only TMEM16A apo representative) and structures with either large terminal truncations (PDB IDs 6BGI, 6BGJ, 8BC1), or largely unmodeled C-terminal domains (PDB ID 6QPI). We further narrowed our final set of structures by selecting the higher resolution of structures sharing similar backbone conformations and have the same number of $Ca^{2+}$ ions bound (*Appendix 1—table 2*). One structure from TMEM16A (PDB ID 8QZC) and TMEM16F (PDB ID 6P46) were excluded because they share a similar conformation to other structures of the same homolog but differ in the number of bound $Ca^{2+}$ ions at the orthosteric site and by a slight elevation of the TM6 C-terminus. In total, we selected 5 nhTMEM16, 2 afTMEM16, 2 TMEM16K, and 11 TMEM16F dimers. We also included 2 cryo-EM structures of TMEM16A. We also chose to simulate a computationally predicted scrambling competent TMEM16F structure based on PDB ID 6QP6 (*Khelashvili et al., 2022*) which has a dilated groove similar to nhTMEM16, afTMEM16, and TMEM16K. We simulated several computationally predicted conductive states of TMEM16A: one based on $Ca^{2+}$-bound TMEM16A (PDB ID 5OYB; *Jia and Chen, 2021*) and three based on simulations of 1PBC-bound TMEM16A (PDB ID 7ZK3) after removing 1PBC from the pore which have significant changes to TM3 and TM4 compared to their experimentally determined starting structures. Several new nhTMEM16 structures were recently published by *Feng et al., 2024a* but not released until after completion of our simulation work and were therefore not included.

##### Atomistic simulation details

TMEM16A atomistic simulations were initiated from a $Ca^{2+}$/1PBC-bound structure (PDB ID 7ZK3) after removal of the inhibitor. The missing residues 260-266, 467-482, 526-527, 669-682 were built and refined using MODELLER (version 10.2; *Sali and Blundell, 1993*), see more details above. Simulations were performed with GROMACS version 2020.6 (*Abraham et al., 2015*) and the CHARMM36 (*Huang and MacKerell, 2013*) and CHARMM36m (c36m) (*Huang et al., 2017*) force fields for lipids and protein, respectively. PROPKA3 was used to check the protonation state of protein residues. E624 and D405 are both weakly protonatable at neutral pH, but likely well solvated and therefore left in their negatively charged states (*Olsson et al., 2011*). The protein was embedded in a 155x155 $Å^2$ POPC bilayer and solvated in 150 mM KCl and the CHARMM TIP3P water model using CHARMM-GUI's Membrane Builder (*Jo et al., 2009*). System charges were neutralized using the same ions. During minimization, equilibration, and production distance restraints with 418.4 kJ $mol^{-1}$ $nm^{-2}$ force constants were applied between the Ca atoms of residues 465 and 489, 454 and 566, 169 and 278, 126 and 176, 196 and 189, 123 and 282, 185 and 200 to stabilize the cytosolic domain. Simulations were run using a 2 fs time step in an NPT ensemble. Temperature was kept at 303.15 K using the Nosé-Hoover (*Evans and Holian, 1985*) thermostat ($\tau_T$ = 1 ps). The pressure of the system was semi-isotropically coupled to a 1 bar reference pressure by the Berendsen (*Berendsen et al., 1984*) and parrinello-rahman (*Parrinello and Rahman, 1981*) barostat ($\tau_P$ = 5 ps, compressibility = 4.5x$10^{-5}$) for equilibration and production respectively. All bonds to H were constrained by the LINCS algorithm (*Hess et al., 1997*). Particle mesh Ewald (*Darden et al., 1993*)-calculated electrostatic and Van der Waals (VdW) interactions were cut off at 1.2 nm. A Verlet cut-off scheme was used for non-bonded interactions. VdW interactions were smoothly switched to zero between 1.0 and 1.2 nm. The protein with its bound $Ca^{2+}$ ions, the membrane, and solvent bath were treated as separate groups for the thermostat coupling and center of mass removal. Harmonic restraints of the protein backbone, sidechains, lipids and dihedrals were applied and slowly reduced over 8 equilibration steps totaling ~32 ns. The equilibrated box size was ~143x143x148 $Å^3$. An identical simulation protocol was used for our atomistic simulations of TMEM16F (6QP6) but used Gromacs version 2018.7. Missing residues 84-88, 143-206, 225-228, 428-444, 490-505, 588-590, 641-644, and 792-794 were modeled using MODELLER (version 10.2; *Sali and Blundell, 1993*). Simulation details for TMEM16A performed by the Chen group are detailed in *Jia and Chen, 2021*. Simulation details for atomistic simulations of TMEM16F (6QP6) performed by the Weinstein group are detailed in *Khelashvili et al., 2022*. Simulation details for atomistic simulations for nhTMEM16 are detailed in *Bethel and Grabe, 2016*.

## Simulated TMEM16A structure selection

TM4/TM5/TM6 residue pair distances from aggregate atomistic trajectories of TMEM16A were submitted to time-independent component analysis (tICA) (*Pérez-Hernández et al., 2013*) and subsequent K-medoids clustering. 100 clusters were then used to construct a Markov-state model and microstates were grouped into macrostates using the improved Perron-cluster cluster analysis (PCCA+) method (*Deuflhard and Weber, 2005*). The above analysis was performed using MSMBuilder (*Harrigan et al., 2017*). We selected the medoids from three of these macrostates (cluster 6, 8, and 10) which are more dilated than the starting structure and predicted to represent different conductive states (*Appendix 1—figure 1*).

## Additional simulation analysis methods

### Maximum density path calculations

One-dimensional paths through the lipid densities were calculated by first selecting a single grid cell near the center of the box, totaling the density of all cells within a 4.7 Å cutoff and then repeating this step for each cell within a 9.4 Å of the first, until a maximum density total was identified. We then saved the centroid of this final set of grid cells and repeated the search using this centroid, or node, as the starting position. This process was continued until either the path length (sum of distances between subsequent nodes) reached 80 Å for lipids (30 Å for water), no new nodes were found, or the next selected node caused the path to deviate sharply (<90°). We only included cells with densities ≥0.0005 for lipids or ≥0.002 for water for this calculation. Path nodes were then interpolated using a B-spline representation in Scipy methods (*Virtanen et al., 2020*) and final nodes were selected from this path to given 0.5 Å gaps between nodes for the lipid paths and 4.7 Å for water.

### Water and ion permeation analysis

For each simulation the positions of water and ion CG beads within a 9.4 Å search radius of the water maximum density pathway were tracked overtime using a custom script that includes MDAnalysis methods (*Michaud Agrawal et al., 2011*, *Gowers et al., 2016*). Water beads were assigned to path nodes if they fell within the bounds of a cylindrical disc (4.7 Å height, 9.4 Å diameter) centered on the bead with its face normal defined by the vector between the current and subsequent node. Permeation events were counted if a water or ion left the search radius into the bath opposite to the one from which it entered the path. The maximum density path was mirrored symmetrically on both subunits and each pathway was tracked independently. We also calculated the flux of water between nodes by counting the net number of waters entering and leaving a cylinder (of the same proportions above) centered on each path node at each 1 ns timestep.

### Diffusion coefficient calculations

Diffusion coefficients for lipids were calculated by first making a 3D interpolated path for each scrambling event and then measuring the minimum path distance to the starting position at each time point. We used a linear least squared regression to fit the slope of each squared displacement curve (squared minimum path distance) and divided the slope by 2 (for 1D diffusion) to obtain the final diffusion coefficient for each lipid. We wrote custom scripts for this analysis using MDAnalysis (*Michaud Agrawal et al., 2011*; *Gowers et al., 2016*) and Scipy methods (*Virtanen et al., 2020*).

### PMF calculations

To calculate the PMFs for each pathway, we assigned each cell from the average PC density to its closest respective mean-lipid pathway node. We then summed the density of all cells assigned to a given node and divided by the total volume of those cells. We used the following equation to convert the density values to energies or PMF.

$$E = \frac{-R * T * \ln\left(\rho/\rho_0\right)}{1000} \tag{1}$$

$\rho_0$ is the density of PC in a 10 µs protein-free membrane simulation, $R$ is the gas constant.

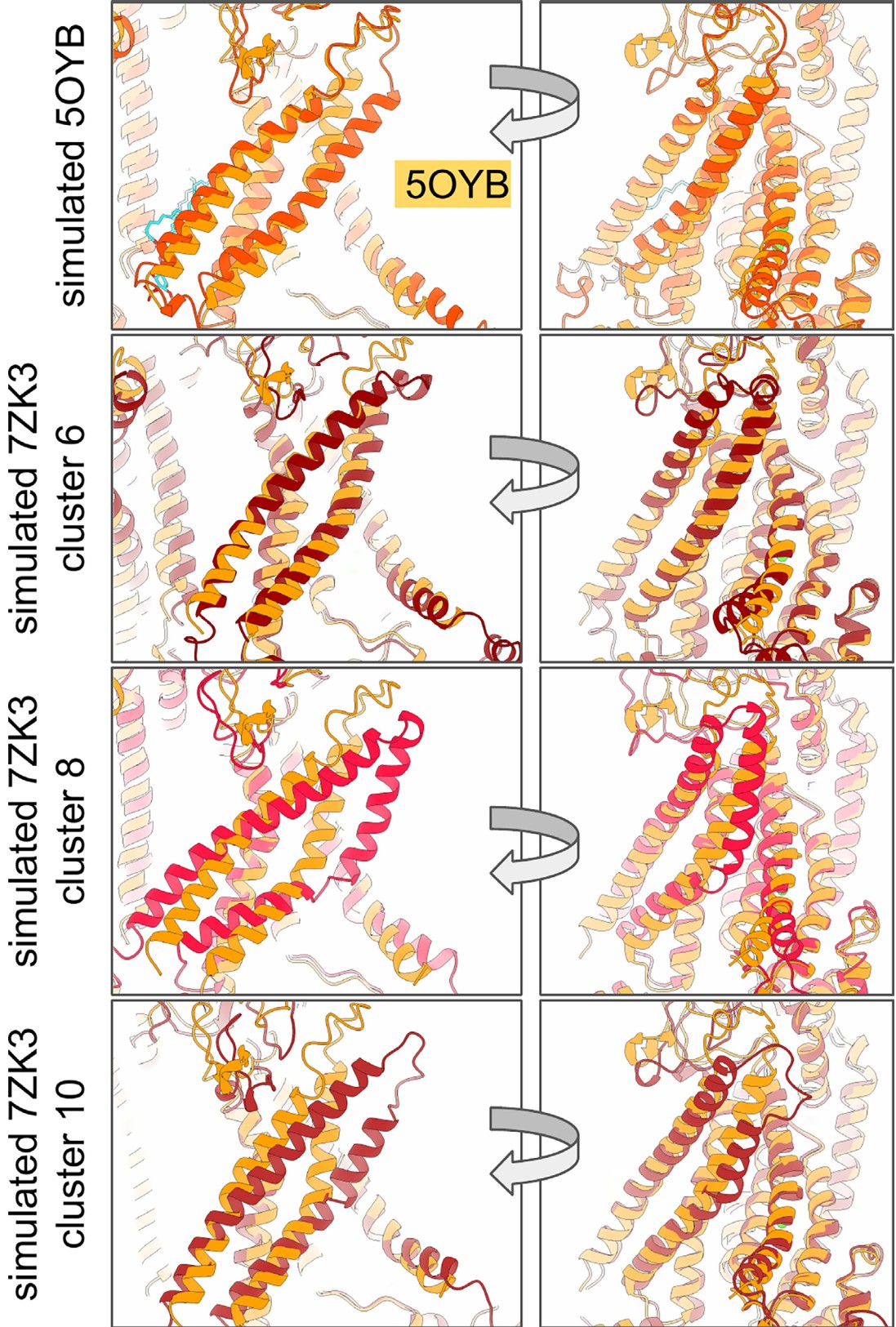

**Appendix 1—figure 1.** Predicted alternative and ion conductive states of TMEM16A. Starting coordinates for simulated conductive states of TMEM16A generated from atomistic MD simulations (darker colors). Each predicted open subunit is aligned to a $Ca^{2+}$-bound TMEM16A cryo-EM structure (PDB ID 5OYB, light orange).

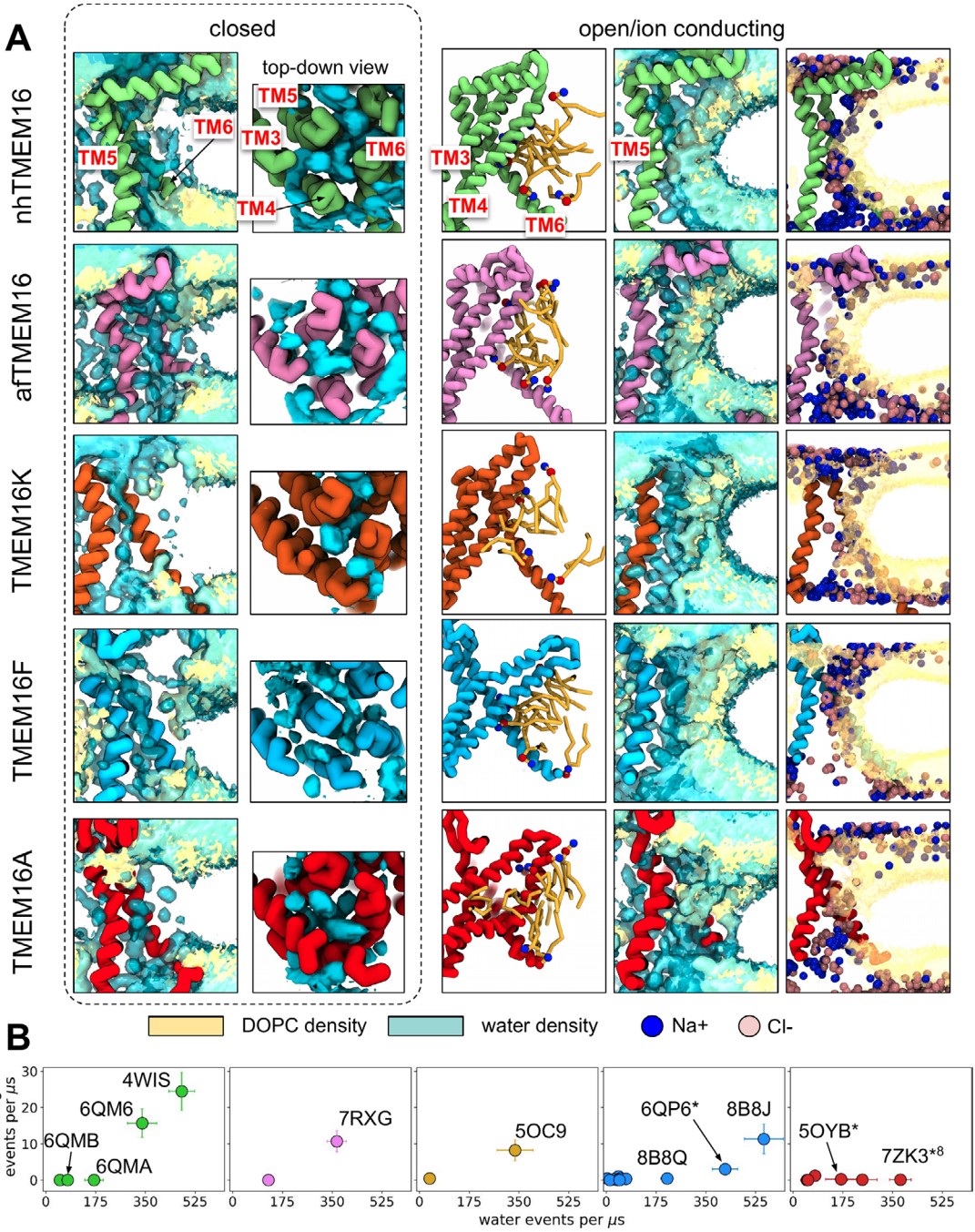

**Appendix 1—figure 2.** Simulated lipid scrambling correlates with water permeation through the TM4/TM6 groove. (**A**) Water and lipid headgroup density in CG simulations of closed (left) and open/ion conducting (right) structures: nhTMEM16 (PDB IDs 6QM4 (closed) and 4WIS (open), green), afTMEM16 (PDB IDs 7RXB (closed) and 7RXG (open), violet), TMEM16K (PDB IDs 6R7X (closed) and 5OC9 (open), gold), TMEM16F (PDB IDs 6QPB (closed) and 8B8J (open), blue) and TMEM16A (PDB ID 5OYG and simulated 7ZK3*[8]). Ions positions (cyan and green beads) shown every 100 frames. TM4 not shown in the first and two rightmost columns. Density is shown at the same isovalue contour for all images. (**B**) The number of scrambling events plotted against the average number of water permeation events per ms. Due to the presence of asymmetric structures the maximum water permeation rate of the two subunits is shown. Water beads were tracked within a 9.4 Å radius of the maximum density pathway (see Appendix 1 - Methods). Density is shown at the same isovalue contour for all images.

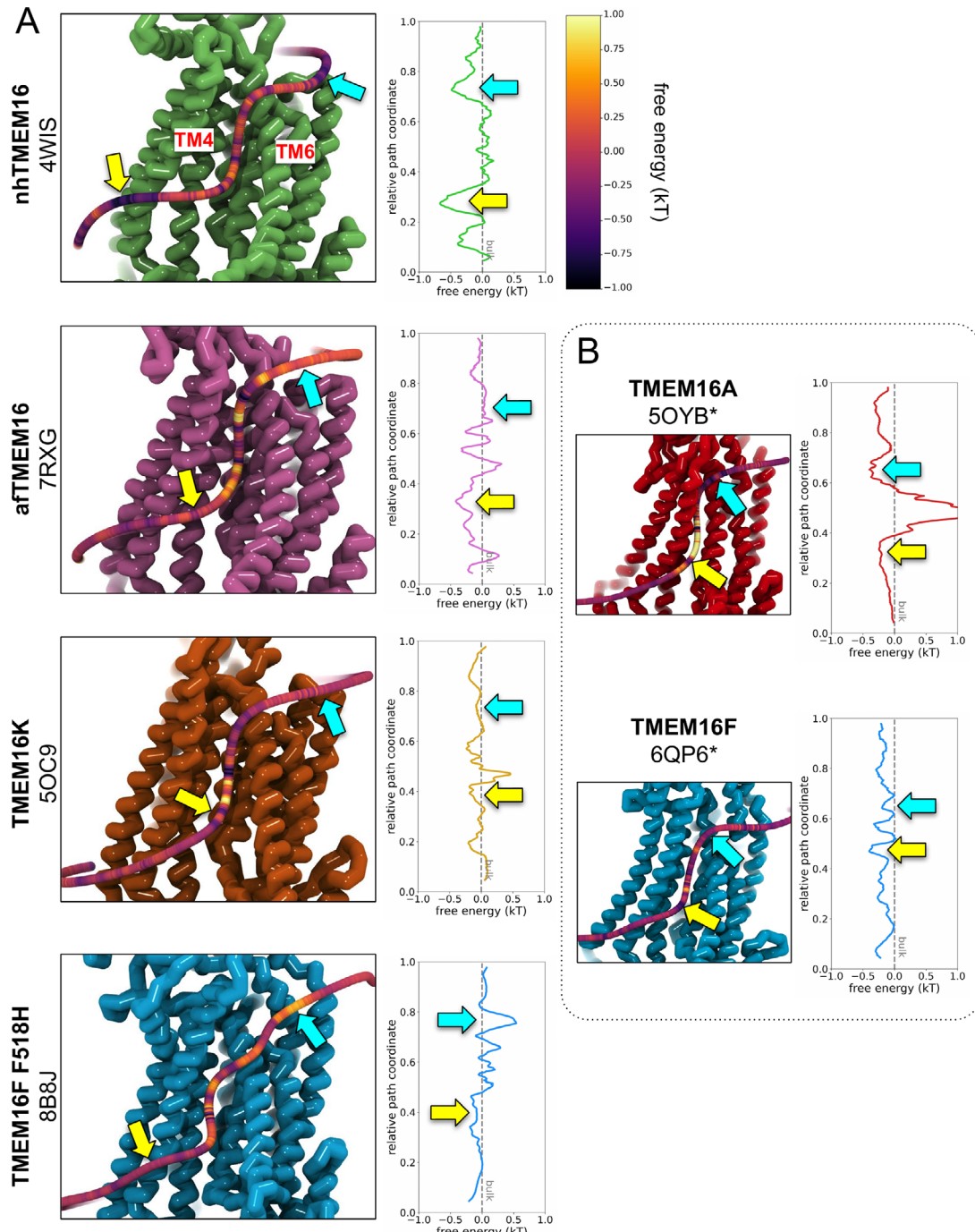

**Appendix 1—figure 3.** PMF estimates from DOPC headgroup density. (**A**) Left: Snapshots of nhTMEM16 (PDB ID 4WIS, green), afTMEM16 (PDB ID 7RXG, violet), TMEM16K (PDB ID 5OC9, orange), and TMEM16F F518H (PDB ID 8B8J, blue), overlaid with a 3D maximum density path (see Methods) calculated from DOPC headgroup positions averaged from both subunits, except for asymmetric TMEM16K and WT TMEM16F paths on subunit with higher scrambling events. Each path is colored by the free energy (kT) calculated from the total density within 4.7 Å of a given node on the path normalized by the lipid headgroup density from simulation of a protein-free DOPC bilayer. Right: the same PMF plotted against its relative position along the maximum density pathway. Colored arrows indicate the same locations along the path. (**B**) PMF-colored 3D maximum lipid density paths for simulated ion conducting TMEM16A (5OYB*, red) and a simulation open WT TMEM16F (6QP6*, blue) simulation with corresponding PMF curve (right).

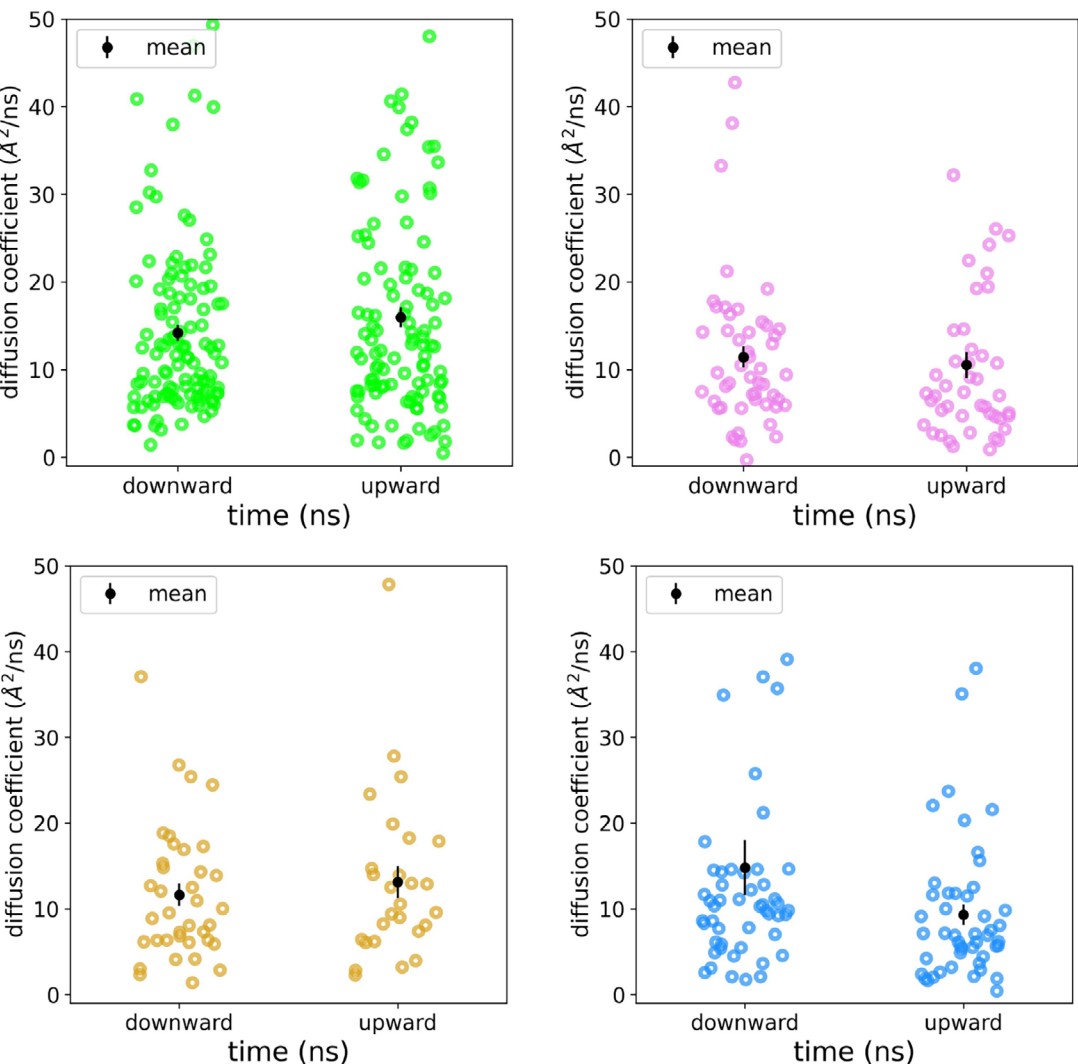

**Appendix 1—figure 4.** Diffusion coefficients for scrambling lipids. Diffusion coefficients were fit to individual mean squared displacement curves for individual DOPC lipids during their scrambling event in simulations of nhTMEM16 (PDB ID 4WIS, green), afTMEM16 (PDB ID 7RXG, violet), TMEM16K (PDB ID 5OC9, gold), and TMEM16F (8B8J, blue). The mean with standard deviation shown in black.

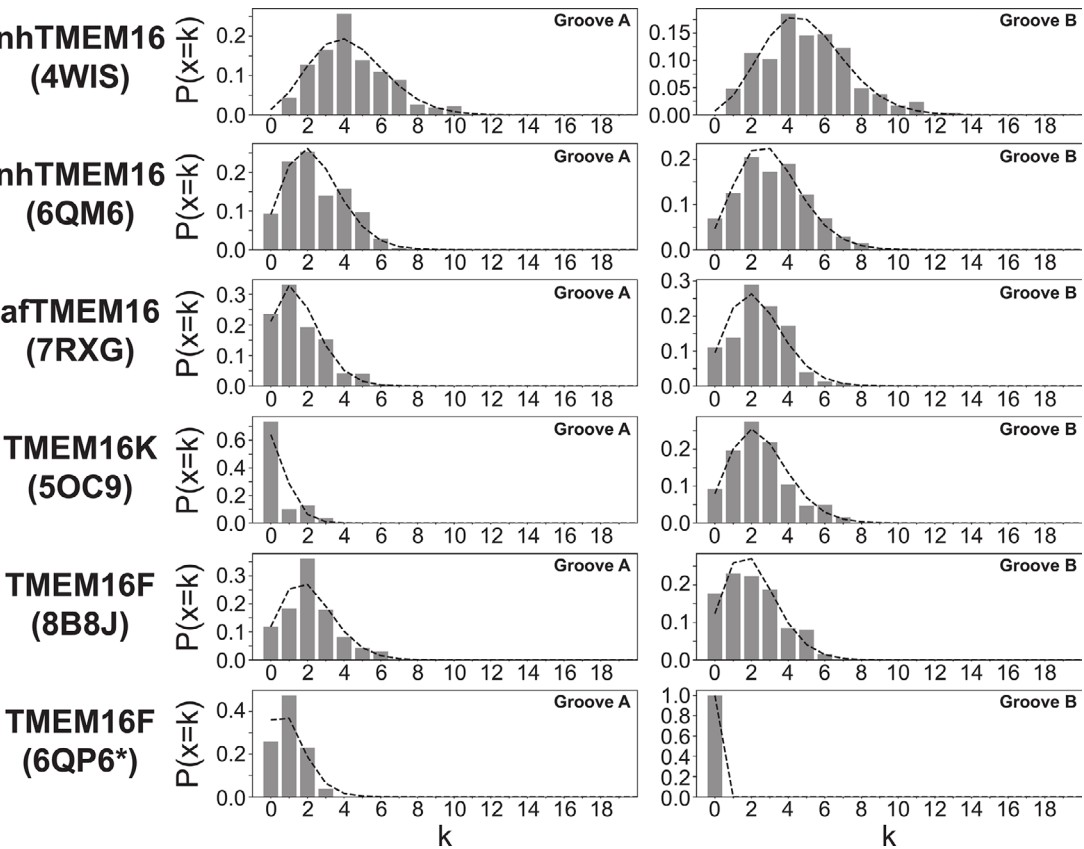

**Appendix 1—figure 5.** Scrambling events are Poisson distributed. Poisson distributions calculated on 400 ns intervals for all scrambling events in groove A (left) and groove B (right) of scrambling-competent structures. Bars represent data, dashed line is the theoretical distribution P(x=k) = $\lambda^k e^{-\lambda}/k!$, with $\lambda$ being the average number of events in the 400 ns interval.

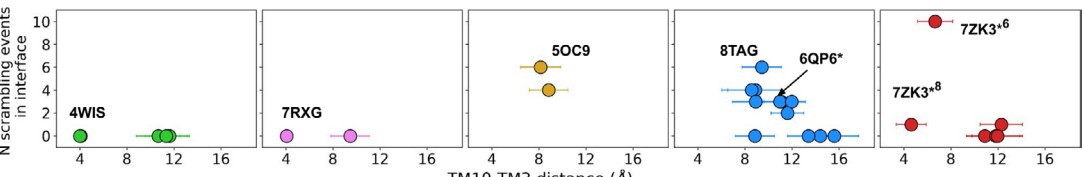

**Appendix 1—figure 6.** 'Out-of-the-groove' scrambling events outside of the dimer interface. (**A**) Snapshots over time of a single lipid scrambling event by TMEM6F over the closed TM4/TM6 groove. (**B**) Snapshots over time of two concurrent scrambling events by TMEM16F over TM6 and TM8. (**C**) Snapshots over time of a single lipid scrambling event by nhTMEM6 over TM3 and TM4. (**D**) Snapshots over time of a single lipid scrambling event by TMEM6A over TM3 and TM4.

**Appendix 1—figure 7.** Number of scrambling events and the width of the dimer interface inner leaflet entrance. The number of scrambling events at the dimer interface plotted against the average minimum distance between TM10 and TM3 on the opposite helix near the inner leaflet-solvent boundary.

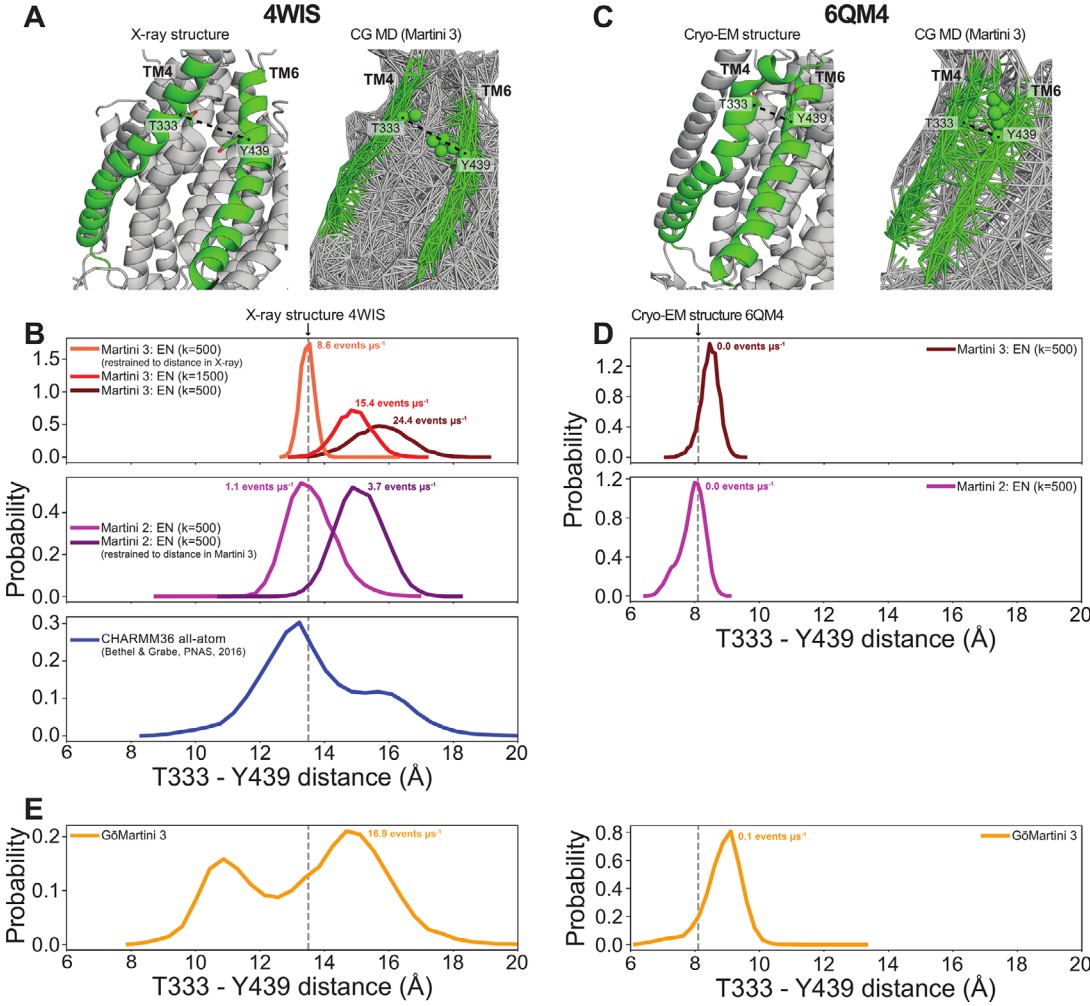

**Appendix 1—figure 8.** TM4/TM6 groove distances vary between different force fields. (**A**) The X-ray structure (left) and the full elastic network for the last frame of our CG MD Martini 3 simulation (right) of open-groove nhTMEM16 (4WIS). TM4 and TM6 are colored green. The dashed line indicates the distance between the Cα-atoms/backbone beads of T333 and Y439, which is used to measure TM4/TM6 distance in B and E. (**B**) Probability distributions of the T333-Y439 distance in 4WIS for different settings of the Martini 3, Martini 2, and CHARMM36 force fields. EN = elastic network. k indicates the force constant used in the EN. Scrambling rates are calculated as described throughout the current work. The CHARMM36 data was taken from *Bethel and Grabe, 2016*. Dashed gray line indicates the distance in the X-ray structure. (**C**) The cryo-EM structure (left) and the full elastic network for the last frame of our CG MD Martini 3 simulation (right) of closed-groove nhTMEM16 (6QM4). TM4 and TM6 are colored green. The dashed line indicates the distance between the Cα-atoms/backbone beads of T333 and Y439, which is used to measure TM4/TM6 distance in D-E. Note the elastic network connections between TM4 and TM6 in green, that are absent in A. (**D**) Probability distributions of the T333-Y439 distance in 6QM4 for different settings of the Martini 3 and Martini 2 force fields. Dashed gray line indicates the distance in the X-ray structure. (**E**) Probability distributions of the T333-Y439 distance in 4WIS (left) and 6QM4 (right) for simulations with the GōMartini 3 force field (*Souza et al., 2024*). Dashed gray line indicates the distance in the respective experimental structures

**Appendix 1—table 1.** TMEM16 structures selected for molecular dynamics (MD) simulations, the concentrations they were solved/constructed in, and modeled missing residues.

| homolog | PDB ID | structure method | detergent/ lipidenvironment | activator/ modulator | organism | modeled loops |
|---|---|---|---|---|---|---|
| nhTMEM16 | 4WIS | X-ray | n-dodecyl-β-D-maltopyranoside (DDM, Anatrace) | 3 mM CaCl$_2$ | fusarium vanettenii 77-13-4 (fungus) | 128–140,465-482,586–593,657-659,685–691 |

*Appendix 1—table 1 Continued on next page*

*Appendix 1—table 1 Continued*

| homolog | PDB ID | structure method | detergent/ lipidenvironment | activator/ modulator | organism | modeled loops |
|---|---|---|---|---|---|---|
| nhTMEM16 | 4WIS | X-ray+Amber/c36m | POPC | 2 Ca$^{2+}$ per chain | fusarium vanettenii 77-13-4 (fungus) | 128–140,465-482,586–593,657-659,685–691, |
| nhTMEM16 | 6QM4 | Cryo-EM | MSP2N2 nanodisc with POPC:POPG = 7:3 | No | fusarium vanettenii 77-13-4 (fungus) | 1–14,417-424,586–594,651-664,685–687 |
| nhTMEM16 | 6QM6 | Cryo-EM | DDM (Anatrace) | No | fusarium vanettenii 77-13-4 (fungus) | 1–14,416-418,468–476,588-594,653–663,685-687 |
| nhTMEM16 | 6QMA | Cryo-EM | MSP2N2 nanodisc with POPC:POPG = 7:3 | 0.3 mM CaCl$_2$ | fusarium vanettenii 77-13-4 (fungus) | 1–14,417-424,469–476,586-594,651–664,685-687 |
| nhTMEM16 | 6QMB | Cryo-EM | MSP2N2 nanodisc with POPC:POPG = 7:3 | 0.3 mM CaCl$_2$ | fusarium vanettenii 77-13-4 (fungus) | 1–14,417-424,468–476,586-594,651–664,685-687 |
| afTMEM16 | 7RXB | Cryo-EM | nanodisc with DOPC:DOPG = 7:3 | No | Aspergillus fumigatus (fungus) | 1–10,25-31,90–102,256-268,312–317,443-463,594–603 |
| afTMEM16 | 7RXG | Cryo-EM | MSP1E3 nanodisc with DOPC:DOPG = 7:3 | 0.5 mM CaCl$_2$ | Aspergillus fumigatus (fungus) | 1–11,725-735 |
| TMEM16K | 5 OC9 | X-ray, LCP | 1-(7Z-hexadecenoyl)-rac-glycerol | 0.1 M CaCl$_2$ | *Homo sapiens* (mammal) | ASYMMETRIC 1–13 A,57–66 A,188–191 A,350–351 A,472–474 A,1-13B,188-191B,348-353B |
| TMEM16K | 6 R7X | Cryo-EM | undecyl maltoside and chloestryl hemisuccinate | 2 mM CaCl$_2$ | *Homo sapiens* (mammal) | 1–12 |
| TMEM16F | 6 P47 | Cryo-EM | digitonin | 0.5 mM CaCl$_2$ | *Mus musculus* (mammal) | 82–88,197-203,257–258,431-444,490–498,639-649 |
| TMEM16F | 6 P48 | Cryo-EM | MSP2N2 nanodisc with POPC:POPE:POPS = 3:1:1 | 2 mM Ca$^{2+}$, PIP$_2$ (1 PIP$_2$ per 50 lipid molecules) | *Mus musculus* (mammal) | 77–89,131-133,225–231 |
| TMEM16F | 6QPB | Cryo-EM | digitonin | No | *Mus musculus* (mammal) | 80–89,491-501,633–645,791-792 |
| TMEM16F | 6QP6 | Cryo-EM | digitonin | 1 mM CaCl$_2$ | *Mus musculus* (mammal) | 82–86,197-201,224–227,489-502,588–590,641-644,792–794 |
| TMEM16F | 6QPC | Cryo-EM | MSP2N2 with POPC:POPG = 3:1 | 1 mM CaCl$_2$ | *Mus musculus* (mammal) | 108–114,258-263,639–646,765-771 |
| TMEM16F | 8TAG | Cryo-EM | MSP2N2-SoyPC nanodisc | 4 mM CaCl$_2$, PIP$_2$ (PIP$_2$:TMEM16 monomer = 4:1) | *Mus musculus* (mammal) | ASYMMETRIC 52–58 A, 83–90 A, 223–231 A, 639–644 A, 490-502B, 869-871B |
| TMEM16F | 8B8G | Cryo-EM | digitonin | No | *Mus musculus* (mammal) | 80–89,491-501,637–645,790-794 |

*Appendix 1—table 1 Continued on next page*

*Appendix 1—table 1 Continued*

| homolog | PDB ID | structure method | detergent/ lipidenvironment | activator/ modulator | organism | modeled loops |
|---------|--------|------------------|------------------------------|----------------------|----------|---------------|
| TMEM16F | 8B8J | Cryo-EM | digitonin | 2 mM CaCl$_2$, 0.01 mM diC8-PI$_{(4,5)}$P$_2$ | *Mus musculus* (mammal) | 82–88,120,221-232,489–502,641-644,792–794 |
| TMEM16F | 8B8Q | Cryo-EM | MSP2N2 nanodisc with POPC:POPG = 3:1 | 2 mM CaCl$_2$ | *Mus musculus* (mammal) | 82–88,120,221-232,639–644,789-794 |
| TMEM16F | 8B8K | Cryo-EM | digitonin | 2 mM CaCl$_2$, 0.01 mM diC8-PI$_{(4,5)}$P$_2$ | *Mus musculus* (mammal) | 82–86,197-201,224–227,489-502,588–590,641-644,792–794 |
| TMEM16F | 8BC0 | Cryo-EM | glyo-dendrimer (GDN, Anatrace) | 2 mM CaCl$_2$ | *Mus musculus* (mammal) | ASYMMETRIC 79–90 A, 220–232 A,254–263 A,588–590 A,640–644 A, 792–794 A, 79-90B,120B,220-232B,254-263B,639-644B |
| TMEM16F | 6QP6 | Cryo-EM+OpenMM + c36 m | POPE:POPG = 7:3 | 1 Ca$^{2+}$per chain, plus 1 Ca2+at dimer interface | *Mus musculus* (mammal) | 1–42, 150–186, 428–444, 489–502, 876–911 |
| TMEM16A | 5OYG | Cryo-EM | digitonin | No | *Mus musculus* (mammal) | 260–266 |
| TMEM16A | 5OYB | Cryo-EM | digitonin | 0.5 mM CaCl$_2$ | *Mus musculus* (mammal) | 260–266,669-682 |
| TMEM16A | 5OYB | Cryo-EM+GROMACS or Amber/c36m | POPC | 2 Ca$^{2+}$ per chain, with 2 docked PIP$_2$ | *Mus musculus* (mammal) | 260–266,467-487,669–682 |
| TMEM16A | 7ZK3 – cluster 6 | Cryo-EM+GROMACS/ c36m | POPC | 3 Ca$^{2+}$ per chain, with 1PBC removed | *Mus musculus* (mammal) | 260–266,467-482,526–527,669-682 |
| TMEM16A | 7ZK3 – cluster 8 | Cryo-EM+GROMACS/ c36m | POPC | 3 Ca$^{2+}$ per chain, with 1PBC removed | *Mus musculus* (mammal) | 260–266,467-482,526–527,669-682 |
| TMEM16A | 7ZK3 – cluster 10 | Cryo-EM+GROMACS/ c36m | POPC | 3 Ca$^{2+}$ per chain, with 1PBC removed | *Mus musculus* (mammal) | 260–266,467-482,526–527,669-682 |

**Appendix 1—table 2.** Experimental structures selected for simulation over other similar structurers.

| Selected Structure (PDB ID) | Similar Structure(s) (PDB ID) |
|------------------------------|-------------------------------|
| 4WIS | 6QM9, 6QM5 |
| 6QMA | 6OY3 |
| 5 OC9 | 6 R65 |
| 7RXB | 7RXA, 7RX3, 6DZ7 |
| 7RXG | 7RX2, 7RWJ, 6E0H, 6E1O |
| 8B8J | 8B8M |
| 8TAG | 8SUR, 8SUN, 8TAI, 8TAL |
| 8BC0 | 8BC1 |
| 6 P48 | 6 P46, 6 P49 |
| 5OYB | 7B5C, 7B5E |
| 5OYG | 7B5D |

