## [Editor Report · eLife Assessment]

This **important** study provides information on the TMEM16 family of membrane proteins, which play roles in lipid scrambling and ion transport. By simulating 27 structures representing five distinct family members, the authors captured hundreds of lipid scrambling events, offering insights into the mechanisms of lipid translocation and the specific protein regions involved in these processes. While the data on comparison of scrambling competence is **compelling**, the evidence for outside-the-groove scramblase activity without experimental validation is missing and is based on a limited set of observed events.

---

## [Referee Report · Reviewer #1 (Public review)]

Summary:

The manuscript investigates lipid scrambling mechanisms across TMEM16 family members using coarse-grained molecular dynamics (MD) simulations. While the study presents a statistically rigorous analysis of lipid scrambling events across multiple structures and conformations, several critical issues undermine its novelty, impact, and alignment with experimental observations.

Review on revised version:

The referee notes that the authors, in their response letter, have concurred with most of the concerns originally raised. Specifically, the authors acknowledge the referee's view that the manuscript primarily confirms previously reported findings and does not present a significantly novel advance, particularly regarding the central observation of groove-mediated lipid scrambling in the open Ca²⁺-bound TMEM16 structures. The authors have also acknowledged the potential discrepancies with existing experimental studies and have addressed this point candidly through additional discussion. Furthermore, the referee appreciates that the authors have echoed the concern regarding the limited statistical robustness of the observed scrambling events.

Given that the authors have essentially affirmed the key points raised in the initial review, the referee believes that these acknowledgements reinforce the basis of the original assessment. Therefore, the referee maintains the original opinion that, despite its technical merits and useful discussion made in the revised version, the manuscript does not offer sufficient novelty or mechanistic depth.

---

## [Referee Report · Reviewer #2 (Public review)]

Summary:

Stephens et al. present a comprehensive study of TMEM16-members via coarse-grained MD simulations (CGMD). They particularly focus on the scramblase ability of these proteins and aim to characterize the "energetics of scrambling". Through their simulations, the authors interestingly relate protein conformational states to membrane's thickness and link those to the scrambling ability of TMEM members, measured as the trespassing tendency of lipids across leaflets. They validate their simulation with a direct qualitative comparison with Cryo-EM maps.

Strengths:

The study demonstrates an efficient use of CGMD simulations to explore lipid scrambling across various TMEM16 family members. By leveraging this approach, the authors are able to bypass some of the sampling limitations inherent in all-atom simulations, providing a more comprehensive and high-throughput analysis of lipid scrambling. Their comparison of different protein conformations, including open and closed groove states, presents a detailed exploration of how structural features influence scrambling activity, adding significant value to the field. A key contribution of this study is the finding that groove dilation plays a central role in lipid scrambling. The authors observe that for scrambling-competent TMEM16 structures, there is substantial membrane thinning and groove widening. The open Ca2+-bound nhTMEM16 structure (PDB ID 4WIS) was identified as the fastest scrambler in their simulations, with scrambling rates as high as 24.4 {plus minus} 5.2 events per μs. This structure also shows significant membrane thinning (up to 18 Å), which supports the hypothesis that groove dilation lowers the energetic barrier for lipid translocation, facilitating scrambling.

The study also establishes a correlation between structural features and scrambling competence, though analyses often lack statistical robustness and quantitative comparisons. The simulations differentiate between open and closed conformations of TMEM16 structures, with open-groove structures exhibiting increased scrambling activity, while closed-groove structures do not. This finding aligns with previous research suggesting that the structural dynamics of the groove are critical for scrambling. Furthermore, the authors explore how the physical dimensions of the groove qualitatively correlate with observed scrambling rates. For example, TMEM16K induces increased membrane thinning in its open form, suggesting that membrane properties, along with structural features, play a role in modulating scrambling activity.

Another significant finding is the concept of "out-of-the-groove" scrambling, where lipid translocation occurs outside the protein's groove. This observation introduces the possibility of alternate scrambling mechanisms that do not follow the traditional "credit-card model" of groove-mediated lipid scrambling. In their simulations, the authors note that these out-of-the-groove events predominantly occur at the dimer interface between TM3 and TM10, especially in mammalian TMEM16 structures. While these events were not observed in fungal TMEM16s, they may provide insight into Ca2+-independent scrambling mechanisms, as they do not require groove opening.

Weaknesses:

A significant challenge of the study is the discrepancy between the scrambling rates observed in CGMD simulations and those reported experimentally. Despite the authors' claim that the rates are in line experimentally, the observed differences can mean large energetic discrepancies in describing scrambling (larger than 1kT barrier in reality). For instance, the authors report scrambling rates of 10.7 events per μs for TMEM16F and 24.4 events per μs for nhTMEM16, which are several orders of magnitude faster than experimental rates. While the authors suggest that this discrepancy could be due to the Martini 3 force field's faster diffusion dynamics, this explanation does not fully account for the large difference in rates. A more thorough discussion on how the choice of force field and simulation parameters influence the results, and how these discrepancies can be reconciled with experimental data, would strengthen the conclusions. Likewise, rate calculations in the study are based on 10 μs simulations, while experimental scrambling rates occur over seconds. This timescale discrepancy limits the study's accuracy, as the simulations may not capture rare or slow scrambling events that are observed experimentally and therefore might underestimate the kinetics of scrambling. It's however, important to recognize that it's hard (borderline unachievable) to pinpoint reasonable kinetics for systems like this using the currently available computational power and force field accuracy. The faster diffusion in simulations may lead to overestimated scrambling rates, making the simulation results less comparable to real-world observations. Thus, I would therefore read the findings qualitatively rather than quantitatively. An interesting observation is the asymmetry observed in the scrambling rates of the two monomers. Since MARTINI is known to be limited in correctly sampling protein dynamics, the authors, in order to preserve the fold, have applied a strong (500 kJ mol-1 nm-2) elastic network. However, I am wondering how the ENM applies across the dimer and if any asymmetry can be noticed in the application of restraints for each monomer and at the dimer interface. How can this have potentially biased the asymmetry in the scrambling rates observed between the monomers? Is this artificially obtained from restraining the initial structure, or is the asymmetry somehow gatekeeping the scrambling mechanism to occur majorly across a single monomer? Answering this question would have far-reaching implications to better describe the mechanism of scrambling.

Notably, the manuscript does not explore the impact of membrane composition on scrambling rates. While the authors use a specific lipid composition (DOPC) in their simulations, they acknowledge that membrane composition can influence scrambling activity. However, the study does not explore how different lipids or membrane environments or varying membrane curvature and tension, could alter scrambling behaviour. I appreciate that this might have been beyond the scope of this particular paper and the authors plan to further chase these questions, as this work sets a strong protocol for this study. Contextualizing scrambling in the context of membrane composition is particularly relevant since the authors note that TMEM16K's scrambling rate increases tenfold in thinner membranes, suggesting that lipid-specific or membrane-thickness-dependent effects could play a role.

Comments on revisions:

I have carefully reviewed the replies of the author, which address the points I raised and improved the manuscript by making the changes outlined in their response. Particularly, I am pleased to see that the authors report ensemble averages in Figure 1-supplement 1 and add relevant information in a newly created table. I welcome the refinement of the discussion towards a cautionary approach in describing quantitatively the findings of experiments and computations for what concerns scrambling rates. I still feel that proper statistical analysis to compare the distributions in Figure 3-figure supplement 6 would have made the points claimed even stronger, but - at the same time - I do see the points of the authors in commenting the differences between these distributions more qualitatively. Overall, I support the publication of this manuscript, it has been a pleasure to read it.

---

## [Referee Report · Reviewer #3 (Public review)]

Summary:

The paper investigates the TMEM16 family of membrane proteins, which play roles in lipid scrambling and ion transport. A total of 27 experimental structures from five TMEM16 family members were analyzed, including mammalian and fungal homologs (e.g., TMEM16A, TMEM16F, TMEM16K, nhTMEM16, afTMEM16). The identified structures were in both Ca²⁺-bound (open) and Ca²⁺-free (closed) states to compare conformations and were preprocessed (e.g., modeling missing loops) and equilibrated. Coarse-grain simulations were performed in DOPC membranes for 10 microseconds to capture the scrambling events. These events were identified by tracking lipids transitioning between the two membrane leaflets and they analysed correlation between scrambling rates, in addition, structural properties such as groove dilation and membrane thinning were calculated. They report 700 scrambling events across structures and the figure 2 elaborates on how open structures show higher activity, also as expected. The authors also address how structures may require open groove, this and other mechanisms around scrambling is a bit controversial in the field.

Strengths:

The strength of this study emerges from comparative analysis of multiple structural starting points and understand global/local motions of the protein with respect to lipid movement. Although the protein is well-studied, both experimentally and computationally, the understanding of conformational events in different family members, especially membrane thickness less compared to fungal scramblases offers good insights.

Weaknesses:

The weakness of the work is to fully reconcile with experimental evidence of Ca²⁺-independent scrambling rates observed in prior studies, but this part is also challenging using coarse-grain molecular simulations. Previous reports have identified lipid crossing, packing defects and other associated events, so it is difficult to place this paper in that context. However, the absence of validation leaves certain claims, like alternative scrambling pathways, speculative.

---

## [Author Response]

The following is the authors’ response to the current reviews.

We wanted to clarify Reviewer #1’s latest comment in the last round of review,

“*Furthermore, the referee appreciates that the authors have echoed the concern regarding the limited statistical robustness of the observed scrambling events.*” We appreciate the follow up information provided from Reviewer #1 that their comment is *specifically* about the low count alternative pathway events that we view at the dimer interface, and not the statistics of the manuscript overall as they believe that “*the study presents a statistically rigorous analysis of lipid scrambling events across multiple structures and conformations (Reviewer #1)*”.

We agree with the Reviewer and acknowledge that overall our coarse-grained study represents the most comprehensive single manuscript of the entire TMEM16 family to date.

The following is the authors’ response to the original reviews.

**Public Review:**

**Reviewer #1 (Public review):**
Summary:The manuscript investigates lipid scrambling mechanisms across TMEM16 family members using coarse-grained molecular dynamics (MD) simulations. While the study presents a statistically rigorous analysis of lipid scrambling events across multiple structures and conformations, several critical issues undermine its novelty, impact, and alignment with experimental observations.Critical issues:(1) Lack of Novelty:The phenomenon of lipid scrambling via an open hydrophilic groove is already well-established in the literature, including through atomistic MD simulations. The authors themselves acknowledge this fact in their introduction and discussion. By employing coarse-grained simulations, the study essentially reiterates previously known findings with limited additional mechanistic insight. The repeated observation of scrambling occurring predominantly via the groove does not offer significant advancement beyond prior work.

We agree with the reviewer’s statement regarding the lack of novelty when it comes to our observations of scrambling in the groove of open Ca2+-bound TMEM16 structures. However, we feel that the inclusion of closed structures in this study, which attempts to address the yet unanswered question of how scrambling by TMEM16s occurs in the absence of Ca2+, offers new observations for the field. In our study we specifically address to what extent the induced membrane deformation, which has been theorized to aid lipids cross the bilayer especially in the absence of Ca2+, contributes to the rate of scrambling (see references 36, 59, and 66). There are also several TMEM16F structures solved under activating conditions (bound to Ca2+ and in the presence of PIP2) which feature structural rearrangements to TM6 that may be indicative of an open state (PDB 6P48) and had not been tested in simulations. We show that these structures do not scramble and thereby present evidence against an out-of-the-groove scrambling mechanism for these states. Although we find a handful of examples of lipids being scrambled by Ca2+-free structures of TMEM16 scramblases, none of our simulations suggest that these events are related to the degree of deformation.

(2) Redundancy Across Systems:The manuscript explores multiple TMEM16 family members in activating and non-activating conformations, but the conclusions remain largely confirmatory. The extensive dataset generated through coarse-grained MD simulations primarily reinforces established mechanistic models rather than uncovering fundamentally new insights. The effort, while statistically robust, feels excessive given the incremental nature of the findings.

Again, we agree with the reviewer’s statement that our results largely confirm those published by other groups and our own. We think there is however value in comparing the scrambling competence of these TMEM16 structures in a consistent manner in a single study to reduce inconsistencies that may be introduced by different simulation methods, parameters, environmental variables such as lipid composition as used in other published works of single family members. The consistency across our simulations and high number of observed scrambling events have allowed us to confirm that the mechanism of scrambling is shared by multiple family members and relies most obviously on groove dilation.

(3) Discrepancy with Experimental Observations:The use of coarse-grained simulations introduces inherent limitations in accurately representing lipid scrambling dynamics at the atomistic level. Experimental studies have highlighted nuances in lipid permeation that are not fully captured by coarse-grained models. This discrepancy raises questions about the biological relevance of the reported scrambling events, especially those occurring outside the canonical groove.

We thank the reviewer for bringing up the possible inaccuracies introduced by coarse graining our simulations. This is also a concern for us, and we address this issue extensively in our discussion. As the reviewer pointed out above, our CG simulations have largely confirmed existing evidence in the field which we think speaks well to the transferability of observations from atomistic simulations to the coarse-grained level of detail. We have made both qualitative and quantitative comparisons between atomistic and coarse-grained simulations of nhTMEM16 and TMEM16F (Figure 1, Figure 4-figure supplement 1, Figure 4-figure supplement 5) showing the two methods give similar answers for where lipids interact with the protein, including outside of the canonical groove. We do not dispute the possible discrepancy between our simulations and experiment, but our goal is to share new nuanced ideas for the predicted TMEM16 scrambling mechanism that we hope will be tested by future experimental studies.

(4) Alternative Scrambling Sites:The manuscript reports scrambling events at the dimer-dimer interface as a novel mechanism. While this observation is intriguing, it is not explored in sufficient detail to establish its functional significance. Furthermore, the low frequency of these events (relative to groove-mediated scrambling) suggests they may be artifacts of the simulation model rather than biologically meaningful pathways.

We agree with the reviewer that our observed number of scrambling events in the dimer interface is too low to present it as strong evidence for it being the alternative mechanism for Ca2+-independent scrambling. This will require additional experiments and computational studies which we plan to do in future research. However, we are less certain that these are artifacts of the coarse-grained simulation system as we observed a similar event in an atomistic simulation of TMEM16F.

Conclusion:Overall, while the study is technically sound and presents a large dataset of lipid scrambling events across multiple TMEM16 structures, it falls short in terms of novelty and mechanistic advancement. The findings are largely confirmatory and do not bridge the gap between coarse-grained simulations and experimental observations. Future efforts should focus on resolving these limitations, possibly through atomistic simulations or experimental validation of the alternative scrambling pathways.
**Reviewer #2 (Public review):**
Summary:Stephens et al. present a comprehensive study of TMEM16-members via coarse-grained MD simulations (CGMD). They particularly focus on the scramblase ability of these proteins and aim to characterize the "energetics of scrambling". Through their simulations, the authors interestingly relate protein conformational states to the membrane's thickness and link those to the scrambling ability of TMEM members, measured as the trespassing tendency of lipids across leaflets. They validate their simulation with a direct qualitative comparison with Cryo-EM maps.Strengths:The study demonstrates an efficient use of CGMD simulations to explore lipid scrambling across various TMEM16 family members. By leveraging this approach, the authors are able to bypass some of the sampling limitations inherent in all-atom simulations, providing a more comprehensive and high-throughput analysis of lipid scrambling. Their comparison of different protein conformations, including open and closed groove states, presents a detailed exploration of how structural features influence scrambling activity, adding significant value to the field. A key contribution of this study is the finding that groove dilation plays a central role in lipid scrambling. The authors observe that for scrambling-competent TMEM16 structures, there is substantial membrane thinning and groove widening. The open Ca2+-bound nhTMEM16 structure (PDB ID 4WIS) was identified as the fastest scrambler in their simulations, with scrambling rates as high as 24.4 {plus minus} 5.2 events per μs. This structure also shows significant membrane thinning (up to 18 Å), which supports the hypothesis that groove dilation lowers the energetic barrier for lipid translocation, facilitating scrambling.The study also establishes a correlation between structural features and scrambling competence, though analyses often lack statistical robustness and quantitative comparisons. The simulations differentiate between open and closed conformations of TMEM16 structures, with open-groove structures exhibiting increased scrambling activity, while closed-groove structures do not. This finding aligns with previous research suggesting that the structural dynamics of the groove are critical for scrambling. Furthermore, the authors explore how the physical dimensions of the groove qualitatively correlate with observed scrambling rates. For example, TMEM16K induces increased membrane thinning in its open form, suggesting that membrane properties, along with structural features, play a role in modulating scrambling activity.Another significant finding is the concept of "out-of-the-groove" scrambling, where lipid translocation occurs outside the protein's groove. This observation introduces the possibility of alternate scrambling mechanisms that do not follow the traditional "credit-card model" of groove-mediated lipid scrambling. In their simulations, the authors note that these out-of-the-groove events predominantly occur at the dimer interface between TM3 and TM10, especially in mammalian TMEM16 structures. While these events were not observed in fungal TMEM16s, they may provide insight into Ca2+-independent scrambling mechanisms, as they do not require groove opening.Weaknesses:A significant challenge of the study is the discrepancy between the scrambling rates observed in CGMD simulations and those reported experimentally. Despite the authors' claim that the rates are in line experimentally, the observed differences can mean large energetic discrepancies in describing scrambling (larger than 1kT barrier in reality). For instance, the authors report scrambling rates of 10.7 events per μs for TMEM16F and 24.4 events per μs for nhTMEM16, which are several orders of magnitude faster than experimental rates. While the authors suggest that this discrepancy could be due to the Martini 3 force field's faster diffusion dynamics, this explanation does not fully account for the large difference in rates. A more thorough discussion on how the choice of force field and simulation parameters influence the results, and how these discrepancies can be reconciled with experimental data, would strengthen the conclusions. Likewise, rate calculations in the study are based on 10 μs simulations, while experimental scrambling rates occur over seconds. This timescale discrepancy limits the study's accuracy, as the simulations may not capture rare or slow scrambling events that are observed experimentally and therefore might underestimate the kinetics of scrambling. It's however important to recognize that it's hard (borderline unachievable) to pinpoint reasonable kinetics for systems like this using the currently available computational power and force field accuracy. The faster diffusion in simulations may lead to overestimated scrambling rates, making the simulation results less comparable to real-world observations. Thus, I would therefore read the findings qualitatively rather than quantitatively. An interesting observation is the asymmetry observed in the scrambling rates of the two monomers. Since MARTINI is known to be limited in correctly sampling protein dynamics, the authors - in order to preserve the fold - have applied a strong (500 kJ mol-1 nm-2) elastic network. However, I am wondering how the ENM applies across the dimer and if any asymmetry can be noticed in the application of restraints for each monomer and at the dimer interface. How can this have potentially biased the asymmetry in the scrambling rates observed between the monomers? Is this artificially obtained from restraining the initial structure, or is the asymmetry somehow gatekeeping the scrambling mechanism to occur majorly across a single monomer? Answering this question would have far-reaching implications to better describe the mechanism of scrambling.

The main aim of our computational survey was to directly compare all relevant published TMEM16 structures in both open and closed states using the Martini 3 CGMD force field. Our standardized simulation and analysis protocol allowed us to quantitatively compare scrambling rates across the TMEM16 family, something that has never been done before. We do acknowledge that direct comparison between simulated versus experimental scrambling rates is complicated and is best to be interpreted qualitatively. In line with other reports (e.g., Li et al, PNAS 2024), lipid scrambling in CGMD is 2-3 orders of magnitude faster than typical experimental findings. In the CG simulation field, these increased dynamics due to the smoother energy landscape are a well known phenomenon. In our view, this is a valuable trade-off for being able to capture statistically robust scrambling dynamics and gain mechanistic understanding in the first place, since these are currently challenging to obtain otherwise. For example, with all-atom MD it would have been near-impossible to conclude that groove openness and high scrambling rates are closely related, simply because one would only measure a handful of scrambling events in (at most) a handful of structures.

Considering the elastic network: the reviewer is correct in that the elastic network restrains the overall structure to the experimental conformation. This is necessary because the Martini 3 force field does not accurately model changes in secondary (and tertiary) structure. In fact, by retaining the structural information from the experimental structures, we argue that the elastic network helped us arrive at the conclusion that groove openness is the major contributing factor in determining a protein’s scrambling rate. This is best exemplified by the asymmetric X-ray structure of TMEM16K (5OC9), in which the groove of one subunit is more dilated than the other. In our simulation, this information was stored in the elastic network, yielding a 4x higher rate in the open groove than in the closed groove, within the same trajectory.

Notably, the manuscript does not explore the impact of membrane composition on scrambling rates. While the authors use a specific lipid composition (DOPC) in their simulations, they acknowledge that membrane composition can influence scrambling activity. However, the study does not explore how different lipids or membrane environments or varying membrane curvature and tension, could alter scrambling behaviour. I appreciate that this might have been beyond the scope of this particular paper and the authors plan to further chase these questions, as this work sets a strong protocol for this study. Contextualizing scrambling in the context of membrane composition is particularly relevant since the authors note that TMEM16K's scrambling rate increases tenfold in thinner membranes, suggesting that lipid-specific or membrane-thickness-dependent effects could play a role.

Considering different membrane compositions: for this study, we chose to keep the membranes as simple as possible. We opted for pure DOPC membranes, because it has (1) negligible intrinsic curvature, (2) forms fluid membranes, and (3) was used previously by others (Li et al, PNAS 2024). As mentioned by the reviewer, we believe our current study defines a good, standardized protocol and solid baseline for future efforts looking into the additional effects of membrane composition, tension, and curvature that could all affect TMEM16-mediated lipid scrambling.

**Reviewer #3 (Public review):**
Strengths:The strength of this study emerges from a comparative analysis of multiple structural starting points and understanding global/local motions of the protein with respect to lipid movement. Although the protein is well-studied, both experimentally and computationally, the understanding of conformational events in different family members, especially membrane thickness less compared to fungal scramblases offers good insights.

We appreciate the reviewer recognizing the value of the comparative study. In addition to valuable insights from previous experimental and computational work, we hope to put forward a unifying framework that highlights various TMEM16 structural features and membrane properties that underlie scrambling function.

Weaknesses:The weakness of the work is to fully reconcile with experimental evidence of Ca²⁺-independent scrambling rates observed in prior studies, but this part is also challenging using coarse-grain molecular simulations. Previous reports have identified lipid crossing, packing defects, and other associated events, so it is difficult to place this paper in that context. However, the absence of validation leaves certain claims, like alternative scrambling pathways, speculative.

Answer: It is generally difficult to quantitatively compare bulk measurements of scrambling phenomena with simulation results. The advantage of simulations is to directly observe the transient scrambling events at a spatial and temporal resolution that is currently unattainable for experiments. The current experimental evidence for the precise mechanism of Ca2+-independent scrambling is still under debate. We therefore hope to leverage the strength of MD and statistical rigor of coarse-grained simulations to generate testable hypotheses for further structural, biochemical, and computational studies.

**Recommendations for the authors:**

**Reviewer #1 (Recommendations for the authors):**
The findings are largely confirmatory and do not bridge the gap between coarse-grained simulations and experimental observations. Future efforts should focus on resolving these limitations, possibly through atomistic simulations or experimental validation of the alternative scrambling pathways.

While we agree with what the reviewer may be hinting at regarding limitations of coarse-grained MD simulations, we believe that our study holds much more merit than this comment suggests. We have provided something that has yet to be done in the field: a comprehensive study that directly compares the scrambling rates of multiple TMEM16 family members in different conformations using identical simulation conditions. Our work clearly shows that a sufficiently dilated grooves is the major structural feature that enables robust scrambling for all TMEM16 scramblases members with solved structures. While all TMEM16s cause significant distortion and thinning of the membrane, we assert that the extreme thinning observed around open grooves is significantly enhanced by the lipid scrambling itself as the two leaflets merge through lipid exchange. We saw no evidence that membrane thinning/distortion alone, in the absence of an open groove, could support scrambling at the rates observed under activating conditions or even the low rates observed in Ca2+-independent scrambling. Moreover, our handful of observations of scrambling events outside of the groove, which has not yet been reported in any study, opens an exciting new direction for studying alternative scrambling mechanisms. That said, we are currently following up on many of the observations reported here such as: scrambling events outside the groove, the kinetics of scrambling, the possibility that lipids line the groove of non-scramblers like TMEM16A, etc. This is being done experimentally with our collaborators through site directed mutagenesis and with all-atom MD in our lab. Unfortunately, it is well beyond the scope of the current study to include all of this in the current paper.

**Reviewer #2 (Recommendations for the authors):**
Major comments and questions:(1) Line 214 and Figure 1- Figure Supplement 1: why have you only compared the final frame of the trajectory to the cryo-EM structure? Even if these comparisons are qualitative, they should be representative of the entire trajectory, not a single frame.

We thank the reviewer for this suggestion and replaced the single-frame snapshots in Figure 1-figure supplement 1 for ensemble-averaged head groups densities. The overall agreement between membrane shapes in CGMD and cryo-EM was not affected by this change.

(2) Lines 228-231: You comment 'Residues in this site on nhTMEM16 and TMEMF also seem to play a role in scrambling but the mechanism by which they do so is unclear.' This is something you could attempt to quantify in the simulations by calculating the correlation between scrambling and protein-membrane interactions/contacts in this site. Can you speculate on a mechanism that might be a contributing factor?

We probed the correlation between these residues and scrambling lipids, as suggested by the reviewer, and interestingly not all scrambling lipids interact with these residues. Yet there is strong lipid density in this vicinity (see insets in Figure 1 and Figure 4-figure supplement 2). These observations lead us to suspect these residues impact scrambling indirectly through influencing the conformation of the protein or flexibility and shape of the membrane. This interpretation fits with mutagenesis studies highlighting a role for these residues in scrambling (see refs 59, 62, and 67). Specifically, Falzone et al. 2022 (ref 59) suggested that they may thin the membrane near the groove, but this has not been tested via structure determination and a detailed model of how they impact scrambling is missing. We could address this question with in silico mutations; however, CG simulation is not an appropriate method to study large scale protein dynamics, and AA simulations are likely best, but beyond the scope of this paper.

(3) Lines 240-245 and Figure 1B: This section discusses the coupling between membrane distortions and the sinusoidal curve around the protein, however, Figure 1B only shows snapshots of the membrane distortions. Is it possible to understand how these two collective variables are correlated quantitatively (as opposed to the current qualitative analysis)?

We believe that it may be possible to quantitatively capture these two key features of the membrane, as we did previously with nhTMEM16 using our continuum elasticity-based model of the membrane (Bethel and Grabe 2016). Our model agreed with all atom MD surfaces to within ~1 Å, hence showing good quantitative agreement throughout the entire membrane. However, we doubt that we could distill the essence of our model down to a simple functional relationship between the sinusoidal wave and pinching, which we think the reviewer is asking. Rather, we believe that the large-scale sinusoidal distortion (collective variable 1) and pinching/distortion (collective variable 2) near the groove arise from the interplay of the specific protein surface chemistry for each protein (patterning of polar and non-polar residues) and the membrane. This is why we chose to simply report the distinct patterns that the family members impose on the surrounding membrane, which we think is fascinating. Specifically, Fig. 1B shows that different TMEM16 family members distort the membrane in different ways. Most notably, fungal TMEM16s feature a more pronounced sinusoidal deformation, whereas the mammalian members primarily produce local pinching. Then, in Fig. 3A we show that the thinning at the groove happens in all structures and is more pronounced in open, scrambling-competent conformations. In other words, proteins can show very strong thinning (e.g. TMEM16K, 5OC9) even though the membrane generally remains flat.

(4) Lines 257-258: Authors comment that TMEM16A lacks scramblase activity yet can achieve a fully lipid-lined groove (note the typo - should be lipid-lined, not lipid-line). Is a fully lipid-lined groove a prerequisite for scramblase activity? Are lipid-lined grooves the only requirement for scramblase activity? Could the authors clarify exactly what the prerequisite for scramblase activity is to avoid any confusion; this will be useful for later descriptions (i.e. line 295) where scrambling competence is again referred to. Additionally, the associated figure panel (Figure 1D) shows a snapshot of this finding but lacks any statistical quantifications - is a fully lipid-lined groove a single event? Perhaps the additional analyses, such as the groove-lipid contacts, may be useful here.

The definition of lipid scrambling is that a lipid fully transitions from one membrane leaflet to the other. While a single lipid could transition through the groove on its own, it is well documented in both atomistic and CG MD simulations, that lipid scrambling typically happens through a lipid-lined groove, as shown in Fig. 1A-B. The lipids tend to form strong choline-to-phosphate interactions with nearest neighbors that make this energetically favorable. That said, lipid-lined grooves are not sufficient for robust scrambling, which is what we show in Fig. 1D where the non-scrambler TMEM16A did in fact feature a lipid-lined groove. As suggested, we performed contact analysis and found that residue K645 on TM6 in the middle of the groove contacts lipids in 9.2% of the simulation frames.

To get a better understanding of how populated the TM4-TM6 pathway is with lipids across all simulated structures, we determined for every simulation frame how many headgroup beads resided in the groove. This indicates that the ion-conductive state of TMEM16A (5OYB*, Fig. 1D) only had 1 lipid in the pathway, on average, meaning that the configuration shown Fig. 1D is indeed exceptional. As a reference, our strongest scrambler nhTMEM16 4WIS, had an average of 2.8 lipids in the groove. We added a table containing the means and standard deviations that resulted from this analysis as Figure 1-Table supplement 1.

(5) Lines 295-298 : The scrambling rates of the Ca²⁺-bound and Ca²⁺-free structures fall within overlapping error margins, it becomes difficult to definitively state that Ca²⁺ binding significantly enhances scrambling activity. This undermines the claim that the Ca²⁺-bound structure is the strongest scrambler. The authors should conduct statistical analyses to determine if the difference between the two conditions is statistically significant.

In contrast to the reviewer’s comment, we do not claim that Ca2+-binding itself enhances lipid scrambling. Instead, what we show is that WT structures that are solved in an open confirmation (all of which are Ca2+-bound, except 6QM6) are robust scramblers. For nhTMEM16, we did not observe any scrambling events for the closed-groove proteins, making further statistical analysis redundant.

(6) The authors claim that the scrambling rates derived from their MD simulations are in "excellent agreement" with experimental findings (lines 294-295), despite significant discrepancy between simulated and experimentally measured rates. For example, the simulated rate of 24.4 {plus minus} 5.2 events/µs for the open, Ca²⁺-bound fungal nhTMEM16 (PDB ID 4WIS) corresponds to approximately 24 million events per second, which is vastly higher than experimental rates. Experimental studies have reported scrambling rate constants of ~0.003 s⁻¹ for TMEM16 family members in the absence of Ca²⁺, measured under physiological conditions (https://doi.org/10.1038/s41467-019-11753-1). Even with Ca²⁺ activation, scrambling rates remain several orders of magnitude lower than the rates observed in simulations. Moreover, this highlights a larger problem: lipid scrambling rates occur over timescales that are not captured by these simulations. While the authors elude to these discrepancies (lines 605-606), they should be emphasised in the text, as opposed to the table caption. These should also be reconducted to differences between the membrane compositions of different studies.

We agree with the spirit of the reviewer’s comment, and because of that, we were very careful not to claim that we reproduce experimental scrambling rates, just that the trends (scrambling-competent, or not) are correct. On lines 294-295, we actually said that the scrambling rates in our simulations excellently agree with “the presumed scrambling competence of each experimental structure”, which is true.

As explained extensively in the discussion section of our paper (and by many others), direct comparison between MD (e.g., Martini 3, but also atomistic force fields) dynamics and experimental measurements is challenging. The primary goal of our paper is to quantify and compare the scrambling capacity of different TMEM16 family members and different states, within a CGMD context.

That said, we agree with the reviewer that we may have missed rare or long-timescale events (as is the case in any MD experiment) and added this point to the discussion.

(7) To address these discrepancies, the authors should: (i) emphasize that simulated rates serve as qualitative indicators of scrambling competence rather than absolute values comparable to experimental findings and (ii) discuss potential reasons for the divergence, such as simulation timescale limitations or lipid bilayer compositions that may favor scrambling and force field inaccuracies.

Please see our answer to question 6. Within the context of our CGMD survey, we confidently call our results quantitative. However, we agree with the reviewer that comparison with experimental scrambling rates is qualitative and should be interpreted with caution. To reflect this, we rewrote the first sentence of the relevant paragraph in the discussion section.

(8) Line 310: Can the authors provide a rationale as to why one monomer has a wider groove than the other? Perhaps a contact analysis could be useful. See the comment above about ENM.

The simulation of Ca2+-bound TMEM16K was initiated from an asymmetric X-ray structure in which chain B features a more dilated groove than chain A (PDB 5OC9). The backbones of TM4 and TM6 in the closed groove (A) are close enough together to be directly interconnected by the elastic network. In contrast, TM4 and TM6 in the more dilated subunit (B) are not restricted by the elastic network and, as a consequence, display some “breathing” behavior (Fig. 3B and Fig. 3-Suppl. 6A), giving rise to a ~4x higher scrambling rate. We explicitly added the word “cryo-EM” and the PDB ID to the sentence to emphasize that the asymmetry stems from the original experimental structure.

When answering this question, we also corrected a mislabeled chain identifier which was in the original manuscript ‘chain A’ when it is actually ‘chain B’ in Fig.2-Suppl. 3A.

(9) Line 312: Authors speculate that increased groove width likely accounts for increased scrambling rates. For statistical significance, authors should attempt to correlate scrambling rates and groove width over the simulation period.

The Reviewer is referring to our description of scrambling rates we measured for TMEM16K where we noted that on average the groove with the highest scrambling rate is also on average wider than the opposite subunit which is below 6 Å. We do not suggest that the correlation between scrambling and groove width is continuous, as the Reviewer may have interpreted from our original submission, but we think it is a binary outcome – lipids cannot easily enter narrow grooves (< 6 Å) and hence scrambling can only occur once this threshold is reached at which point it occurs at a near constant rate. We showed this for 4 different family members in the original Fig. 3B, where scrambling events (black dots) were much more likely during, or right after, groove dilation to distances > 6 Å.

(10) Line 359: Authors have plotted the minimum distance between residues TM4 and TM6 in Fig. 3A/B, claiming that a wide groove is required for scrambling. Upon closer examination, it is clear that several of these distributions overlap, reducing the statistical significance of these claims. Statistical tests (i.e. KS-tests) should be performed to determine whether the differences in distributions are significant.

The Reviewer appears to be asking for a statistical test between the six distance distributions represented by the data in Fig. 3A for the scrambling competent structures (6QP6*, 8B8J, 6QM6, 7RXG, 4WIS, 5OC9), and we think this is being asked because it is believed that we are making a claim that the greater the distance, the greater the scrambling rate. If we have interpreted this comment correctly, we are not making this claim. Rather, we are simply stating that we only observe robust scrambling when the groove width regularly separates beyond 6 Å. The full distance distributions can now be found in Figure 3-figure supplement 6B, and we agree there is significant overlap between some of these distributions. However, the distinguishing characteristic of the 6 distributions from scrambling competent proteins is that they all access large distances, while the others do not. Notably, TMEM16F proteins (6QP6*, 8B8J) are below the 6 Å threshold on average, but they have wide standard deviations and spend well over ¼ of their time in the permissive regime (the upper error bar in the whisker plots in Fig. 3A is the 75% boundary).

(11) Line 363-364: The authors state that all TMEM16 structures thin the membrane. Could the authors include a description of how membrane thinning is calculated, for instance, is the entire membrane considered, or is thinning calculated on a membrane patch close to the protein? Do membrane patches closer to the transmembrane protein increase or decrease thickness due to hydrophobic packing interactions? The latter question is of particular concern since Martini3 has been shown to induce local thinning of the membrane close to transmembrane helices, yielding thicknesses 2-3 Å thinner than those reported experimentally (https://doi.org/10.1016/j.cplett.2023.140436). This could be an important consideration in the authors' comparison to the bulk membrane thickness (line 364). Finally, how is the 'bulk membrane thickness' measured (i.e., from the CG simulations, from AA simulations, or from experiments)?

Regarding the calculation of thinning and bulk membrane thickness, as described in Method “Quantification of membrane deformations”, the minimal membrane thickness, or thinning, is defined as the shortest distance between any two points from the interpolated upper and lower leaflet surfaces constructed using the glycerol beads (GL1 and GL2). Bulk membrane thickness is calculated by taking the vertical distance between the averaged glycerol surfaces at the membrane edge.

The concern of localized membrane deformation due to force field artifacts is well-founded. However, the sinusoidal deformations shown here are much greater than 2-3 Å Martini3 imperfections, and they extend for up to 10 Å radially away from the protein into the bulk membrane (see Figure 3-figure supplement 1-5 for more of a description). Most importantly, the sinusoidal wave patterns set up by the proteins is very similar to those described in the previous continuum calculation and all-atom MD for nhTMEM16 (https://www.pnas.org/doi/full/10.1073/pnas.1607574113).

(12) Line 374: The authors state a 'positive correlation' between membrane thinning/groove opening and scrambling rates. To support this claim, the authors should report. the correlation coefficients.

We have removed any discussion concerning correlations between the magnitude of the scrambling rate and the degree of membrane thinning/groove opening. Rather we simply state that opening beyond a threshold distance is required for robust scrambling, as shown in our analysis in Fig. 3A.

Concerning the relation between thinning and scrambling: Instantaneous membrane thinning is poorly defined (because it is governed by fluctuations of single lipids), and therefore difficult to correlate with the timing of individual scrambling events in a meaningful way. Moreover, as we state later in that same section, “we argue that the extremely thin membranes are likely correlated with groove opening, rather than being an independent contributing factor to lipid scrambling”.

(13) Line 396: It is stated that TMEM16A is not a scramblase but the simulating scrambling activity is not zero. How can you be sure that you are monitoring the correct collective variable if you are getting a false positive with respect to experiments?

We only observe 2 scrambling events in 10 ms, which is a very small rate compared to the scrambling competent states. In a previous large survey Martini CG simulation study that inspired our protocol (Li et al, PNAS 2024), they employed a 1 event/ms cut-off to distinguish scramblers from non-scramblers. Hence, they would have called TMEM16A a non-scrambler as well. We expect that false negatives in this context might be an artifact of the CG forcefield, or it could be that TMEM16A can scramble but too slowly to be experimentally detected. Regarding the collective variable for lipid flipping, it is correct, and we know that this lipid actually flipped.

(14) Line 402: Distance distributions for the electrostatic interactions between E633 and K645 should be included in the manuscript. This is also the case for the interactions between E843-K850 (lines 491-492).

Our description of interactions between lipid headgroups and E633 and K645 in TMEM16A (5OYB*) are based on qualitative observations of the MD trajectory, and we highlight an example of this interaction in Figure 3-video 4. The video clearly shows that the lipid headgroups in the center of the groove orient themselves such that the phosphate bead (red) rests just above K645 (blue) and at other times the choline bead (blue) rests just below E633 (red). We do not think an additional plot with the distance distributions between lipids and these residues will add to our understanding of how lipids interact residues in the TMEM16A pore.

We made a similar qualitative observation for the interaction between the POPC choline to E843 and POPC phosphate to K850 while watching the AAMD simulation trajectory of TMEM16F (PDB ID 6QP6). Given that this was a single observation, and the same interactions does not appear in CG simulation of the same structure (see simulation snapshots in Figure 4-figure supplement 5) we do not think additional analysis would add significantly to our understanding of which residues may stabilize lipids in the dimer interface.

(15) Lines 450-451: 'As the groove opens, water is exposed to the membrane core and lipid headgroups insert themselves into the water-filled groove to bridge the leaflets.' Is this a qualitative observation? Could the authors report the correlation between groove dilation and the number of water permeation events?

Yes, this is qualitative, and it sketches the order of events during scrambling, and we revised the main text starting at line 450 to indicate this. As illustrated by the density isosurfaces in Appendix 1-Figure 2A, the amount of water found in the closed versus open grooves is striking – there is a significant flood of water that connects the upper and lower solutions upon groove opening. Moreover, Appendix 1-Figure 2B shows much greater water permeation for open structures (4WIS, 7RXG, 5OC9, 8B8J, …) compared to closed structures (6QMB, 6QMA, 8B8Q, and many of the non-labeled data in the figure that all have closed grooves and near 0 water permeation). A notable exception is TMEM16A (7ZK3*8), which has water permeation but a closed groove and little-to-no lipid scrambling.

Minor Comments:(1) Inconsistent use of '10' and 'ten' throughout.

We like to kindly point out that we do not find examples of inconsistent use.

(2) Line 32: 'TM6 along with 3, 4 and 5...' should be 'TM6 along with TM3, TM4 and TM5...'. Same in line 142. Naming should stay consistent.

Changes are reflected in the updated manuscript.

(3) Line 141: do you mean traverse (i.e. to travel across)? Or transverse (i.e. to extend across the membrane)?

This is a typo. We meant “traverse”. Thanks for pointing it out.

(4) Line 142: 'greasy' should be 'strongly hydrophobic'.

Changes are reflected in the updated manuscript.

(5) Line 143-144: "credit card mechanism" requires quotation marks.

Changes are reflected in the updated manuscript.

(6) Line 144: state if Nectria haematococca is mammalian or fungal, this is not obvious for all readers.

Changes are reflected in the updated manuscript.

(7) Line 147-148: Is TMEM16A/TMEM16K fungal or mammalian? What was the residue before the mutation and which residue is mutated? Perhaps the nomenclature should read as TMEM16X10Y where X=the residue prior to the mutation, 10 is a placeholder for the residue number that is mutated and Y=the new residue following mutation.

“TMEM16” is the protein family. “A” denotes the specific homolog rather than residue.

(8) Lines 157-158: same as 10, it is unclear if these are fungal or mammalian.

Clarifications added.

(9) Line 184: "...CGMD simulation" should be "...CGMD simulations".

Changes made.

(10) Line 191-192: It would help to create a table of all of the mutants (including if they are mammalian or fungal) summarizing the salt concentrations, lipid and detergent environments, the presence of modulators/activators, etc.

We added this information to Appendix 1-Table 1 in the supplemental information. We did not specify NaCl concentrations, because they all experimental procedures used standard physiological values for this (100-150 mM).

(11) Line 210: inconsistencies with 'CG' and 'coarse-grain'.

Changes made.

(12) Figure 1 caption: '...totaling ~2μs (B)...' is missing the fullstop after 2μs.

Changes made.

(13) Figure 1B: it may be useful to label where the Ca2+ ion binds or include a schematic.

We updated Fig. 1A to illustrate where Ca2+ binds.

(14) Line 311: Are these mean distances? The authors should add standard deviations.

Yes, they are. We added the standard deviations to the text.

(15) Line 321-322: Perhaps a schematic in Figure 2 would be useful to visualize the structural features described here.

We would kindly refer interested readers to reference [60].

(16) Line 377: '...are likely a correlate of groove opening...' should read as: '...are likely correlated to groove opening...'.

Thank you for pointing it out. Changes made.

(17) Line 398: the '...empirically determined 6Å threshold for scrambling.' Was this determined from the simulations or from experiments? What does "empirically" mean here? Please state this.

This value was determined from the simulations. Based on our analysis of the correlation between scrambling rate and groove dilation, we found that the minimal TM4/6 distance of 6 Å can distinguish between the high and low activity scramblers. The exact numerical value is somewhat arbitrary as there is a range of values around 6 Å that serve to distinguish scramblers from non-scramblers.

(18) Figure 4: This figure should be labelled as A, B, C and D, with the figure caption updated accordingly.

We updated Figure 4 and its caption.

**Reviewer #3 (Recommendations for Authors):**
The authors must do additional simulations to further validate their claim with different lipids and further substantiate dimer interface independent of Ca2+ ions.

Thank you for the suggestion. We completely agree that studying scrambling in the context of a diverse lipid environment is an exciting area to explore. We are indeed actively working on a project that shares the similar idea. We decided not to include that study because we think the additional discussion involved would be excessive for the current manuscript. We, however, look forward to publishing our findings in a separate manuscript in the near future. In terms of Ca2+-independent scrambling, we are planning with our experimental collaborator for mutagenesis studies that target the residues we identified along the dimer interface.

Since calcium ions are critical for the stability of these structures, authors should show that they were placed throughout the simulations consistently.

As stated in the method section “Coarse-grained system preparation and simulation detail”, all Ca2+ ions are manually placed into the coarse-grained structure from the beginning of the simulation at their identical corresponding position in the experimental structure and harmonically bonded to adjacent acidic residues throughout the duration of simulation. We have also added a label to Fig 1A to indicate where the two Ca2+ ions are located.

The comparison with experimental structures should be consistent with complete simulation, and not the last structure of the trajectory. Depending on the conformational variability, this might be misleading.

We agree and updated Fig. 1-supplement figure 1 accordingly. The overall agreement between membrane shapes in CGMD and cryo-EM was not affected by this change.